# Resolution of R-loops by INO80 promotes DNA replication and maintains cancer cell proliferation and viability

Lisa Prendergast [1,8], Urszula L. McClurg [2,10], Rossitsa Hristova[3,10], Rolando Berlinguer-Palmini [4], Sarah Greener[5], Katie Veitch[5], Inmaculada Hernandez[5,9], Philippe Pasero[6], Daniel Rico [5], Jonathan M. G. Higgins [5], Anastas Gospodinov [3✉] & Manolis Papamichos-Chronakis [2,7✉]

Collisions between the DNA replication machinery and co-transcriptional R-loops can impede DNA synthesis and are a major source of genomic instability in cancer cells. How cancer cells deal with R-loops to proliferate is poorly understood. Here we show that the ATP-dependent chromatin remodelling INO80 complex promotes resolution of R-loops to prevent replication-associated DNA damage in cancer cells. Depletion of INO80 in prostate cancer PC3 cells leads to increased R-loops. Overexpression of the RNA:DNA endonuclease RNAse H1 rescues the DNA synthesis defects and suppresses DNA damage caused by INO80 depletion. R-loops co-localize with and promote recruitment of INO80 to chromatin. Artificial tethering of INO80 to a LacO locus enabled turnover of R-loops in *cis*. Finally, counteracting R-loops by INO80 promotes proliferation and averts DNA damage-induced death in cancer cells. Our work suggests that INO80-dependent resolution of R-loops promotes DNA replication in the presence of transcription, thus enabling unlimited proliferation in cancers.

[1] Cancer Research UK Newcastle Drug Discovery Unit, Newcastle University Centre for Cancer, Newcastle University, Paul O'Gorman Building, Newcastle upon Tyne NE2 4HH, UK. [2] Institute of Systems, Molecular and Integrative Biology, University of Liverpool, Liverpool L69 7ZB, UK. [3] Roumen Tsanev Institute of Molecular Biology, Bulgarian Academy of Sciences, Sofia, Bulgaria. [4] Bio-Imaging Unit, Newcastle University, Newcastle upon Tyne NE2 4HH, UK. [5] Biosciences Institute, Newcastle University, Newcastle upon Tyne NE2 4HH, UK. [6] Institute of Human Genetics, CNRS UMR9002 and University of Montpellier, Equipe Labéllisée Ligue Contre le Cancer, 34090 Montpellier, France. [7] Institute for Cell and Molecular Biosciences, Newcastle University, Newcastle Upon Tyne NE2 4HH, UK. [8] Present address: Drug Discovery Unit, Cancer Research UK Manchester Institute, The University of Manchester, Alderley Park, SK10 4TG Manchester, UK. [9] Present address: CNAG-CRG, Centre for Genomic Regulation (CRG), The Barcelona Institute of Science and Technology, Barcelona 08028, Spain. [10] These authors contributed equally: Urszula L. McClurg, Rossitsa Hristova. ✉email: agg@bio21.bas.bg; manolis@liverpool.ac.uk

In proliferating cells, conflicts between DNA replication and transcription are one of the greatest threats to genome stability. Failure to resolve transcription-replication interference can lead to replication stress, which is characterized by stalling of the replication fork and induction of DNA breaks, with detrimental effects to cell proliferation and homeostasis. Mounting evidence indicates that co-transcriptional RNA:DNA hybrid structures known as R-loops[1] are a major obstacle to replication fork progression. While R-loops play a regulatory role in transcription[1], encounters of forks with R-loops are a potent source of replication stress[2–4] and are particularly genotoxic when they occur in a head-on orientation[5]. Recent studies indicate that R-loops are highly abundant and induce replication stress in cancer cells[6,7]. This raises the question how cancer cells sustain sufficient DNA synthesis rates in the presence of increased transcription-replication conflicts. Several factors have been shown to prevent the formation or promote resolution of R-loops [reviewed in refs. [2,3,8]]. However, the molecular pathways protecting forks from collisions with R-loops in cancer cells are poorly understood.

Emerging evidence indicates a role for chromatin structure in R-loop control. Mutations in the core nucleosomal histones H3 and H4 lead to accumulation of R-loops[9]. The histone chaperone complex FACT, which promotes nucleosomal integrity[10] and facilitates transcription in the presence of chromatin[11], prevents R-loop accumulation and promotes resolution of transcription-replication conflicts[12]. Nevertheless, our understanding of the role of chromatin regulation in R-loop modulation remains critically limited. Importantly, whether the role of chromatin is that to solely suppress R-loop formation, or to also promote R-loop resolution, remains unknown.

The chromatin landscape is shaped by the action of ATP-dependent chromatin remodelling enzymes that alter the structure, composition or position of nucleosomes[13]. The INO80 complex, which contains the INO80 ATPase, is a structurally and functionally evolutionary conserved chromatin remodelling complex[14]. In yeast, INO80 has been shown to facilitate degradation of RNA Polymerase II during replication stress conditions in order to preserve genome stability[15,16]. Mammalian INO80 is required for replication fork stability and recovery following replication stress[17], whilst it also promotes DNA replication in unperturbed conditions[18]. Nevertheless, the role of INO80 in DNA replication remains unclear.

Several INO80 subunits are overexpressed in different cancers such as breast[19], neuroendocrine prostate cancer[20] and melanoma[21], and their expression levels can correlate positively with bad prognosis[21,22]. Depletion of INO80 in cancer cells decreases oncogenic transcription, compromises cell proliferation and subsequent tumour growth[21]. These observations suggest a critical function for INO80 in sustaining cancer development and progression. However, the role of INO80 in cancer cell proliferation remains largely elusive.

Here we investigate the mechanisms that protect cells from replication-associated DNA damage. By using oncogene-driven prostate cancer PC3 cells, we find that INO80 prevents replication stress-induced DNA damage and promotes proper and efficient DNA synthesis by counteracting accumulation of R-loops. INO80 is recruited to R-loop-enriched sites across the genome independently of gene expression levels. Artificial tethering of INO80 at a genomic site enriched in R-loops results in turnover of R-loops in *cis*. Notably, removal of R-loops by overexpression of the RNA:DNA endonuclease RNAse H1 rescues the growth defects caused by INO80 depletion in PC3 cells, NRAS-dependent melanoma WM1361 cells and estrogen-dependent breast cancer MCF7 cells, while inhibition of the BER pathway sensitizes INO80-depleted cancer cells to lethality.

Our results suggest that R-loop resolution driven by INO80 prevents genotoxic collisions between transcription and replication, enabling unlimited proliferation of cancer cells.

## Results

### INO80 promotes DNA replication by counteracting R-loops.
We sought to understand the underlying cause for defective DNA replication in human cells lacking INO80. We questioned whether the role of INO80 in promoting DNA replication is dependent on transcription. siControl and siINO80 PC3 cells were treated with the transcriptional inhibitors α-amanitin or cordycepin and analysed for DNA synthesis rates by CldU/IdU DNA fibre pulse labelling assay (Fig. 1a). INO80 depletion led to significantly decreased DNA synthesis (Fig. 1b, c), as expected[17,18]. Total labelling of DNA fibres indicated that DNA fragmentation was not the cause of reduced fibre length (Supplementary Fig. 1). Ectopic expression of siRNA immune INO80 cDNA rescued the replication defect in siINO80 cells, indicating that the replication defect is specifically due to loss of INO80 (Supplementary Fig. 2a–d). Treatment with transcription inhibitors rescued partially but significantly the DNA synthesis defect of siINO80 cells (Fig. 1b, c and Supplementary Fig. 1a), suggesting that transcription impedes DNA replication in the absence of INO80.

We asked whether R-loops are the cause of the DNA replication defect in siINO80 cells. siControl and siINO80 cells were co-transfected with either an empty vector (EV) or a plasmid overexpressing the endonuclease RNAse H1 (RNAseH1 o/e), which specifically targets and removes RNA:DNA hybrids from the genome, and DNA synthesis rates were evaluated using DNA fibre labelling as before (Fig. 1d–f and Supplementary Fig. 2g, h). RNAse H1 overexpression reduced DNA synthesis in control cells and slightly increased the number of cells in S-phase as indicated by FACS cell cycle profile analysis (Fig. 1e, f and Supplementary Figs. 2h and 3a–c). INO80 depletion led to reduced length of both IdU-stained and CldU-stained DNA fibres and increased accumulation of cells in early S-phase (Fig. 1e, f and Supplementary Fig. 3b–d). RNAse H1 overexpression rescued the DNA synthesis and cell cycle defects of siINO80 cells (Fig. 1e, f and Supplementary Figs. 2 and 3b, c). These data indicate that R-loops are an obstacle to DNA replication in the absence of INO80.

The ACTR8/Arp8 subunit of INO80 is required for the complex's chromatin remodelling activity[23]. Depletion of ACTR8 led to a significant decrease in DNA synthesis rates (Supplementary Fig. 3e–h), while overexpression of RNAse H1 in siACTR8 cells rescued the replication defect (Supplementary Fig. 3g, h). We tested whether chromatin relaxation in cells depleted of INO80 would rescue DNA replication similar to the rescue observed upon RNase H1 overexpression. The histone deacetylase (HDAC) inhibitor SAHA/Vorinostat induces hyperacetylation of the N-terminal tails of histones H3 and H4, creating a more open nucleosomal structure. Addition of Vorinostat in control cells reduced DNA synthesis rates[24], while it significantly rescued the DNA synthesis rates of siINO80 cells (Fig. 1i,). When Vorinostat was added in siINO80 cells overexpressing RNAse H1, it did not further increase DNA synthesis rates compared to untreated siINO80 cells overexpressing RNAse H1 (Fig. 1i). This supports an epistatic functional relationship between RNase H1 and Vorinostat in rescuing DNA synthesis in the absence of INO80. Together these data suggest that chromatin regulation by INO80 counteracts R-loops to promote replication fork progression.

### RNAse H1 overexpression rescues stalled forks in siINO80.
To distinguish between slower fork movement, or increased fork stalling by R-loops in siINO80 cells, we analysed the progression

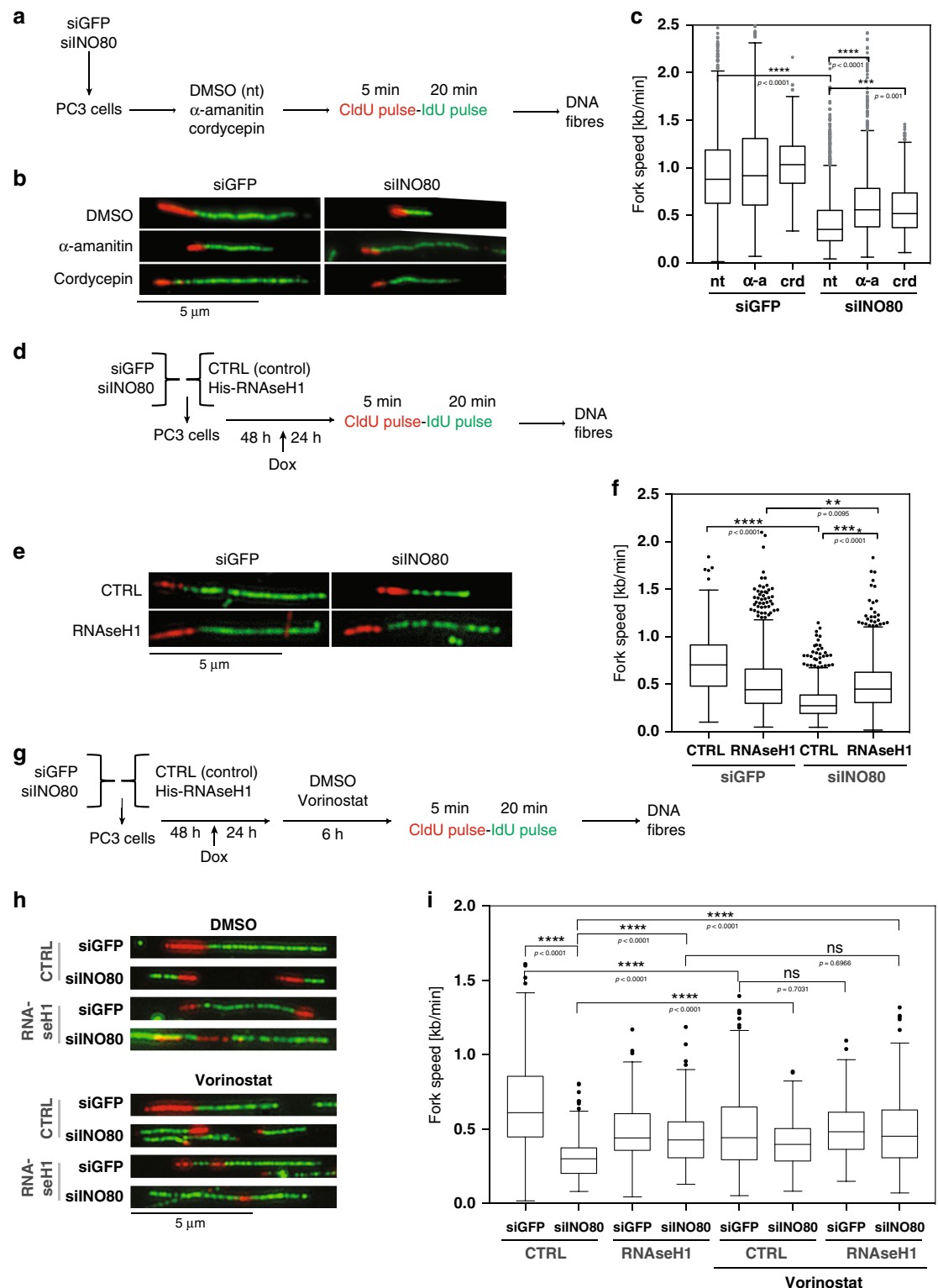

of sister replication forks (Fig. 2a–d). In control cells, the majority of sister forks progressed at a similar rate from a given origin and generated symmetrical patterns of IdU/CldU incorporation (Fig. 2b, c). However, in siINO80 cells, 89% of the forks were asymmetrical with a greater than 2-fold difference in DNA synthesis rates between the two sister forks compared to control cells (Fig. 2b–d). This indicates that forks stall more frequently in cells lacking INO80. ssDNA fibre analysis showed intact DNA fibres at non-symmetrical sister replication forks in siINO80 cells

(Supplementary Fig. 4), ruling out the possibility that the replication fork asymmetry in siINO80 cells is caused by DNA damage. When RNAse H1 was overexpressed in siINO80 cells, fork symmetry was recovered (Fig. 2), strongly suggesting that R-loops cause increased replication fork stalling in the absence of INO80.

**RNAse H1 reduces DNA damage in replicating siINO80 cells.** Collisions between replication forks and R-loops induce DNA

**Fig. 1 R-loops slow replication rate in INO80-depleted cells. a** Schematic representation of the experimental approach. PC3 cells were transfected with esiRNAs against either GFP or INO80. Three days later, cells were treated with α-amanitin (α-a) or cordycepin (crd) for 3 h or left untreated as control and then subjected to fibre labelling analysis. **b** Representative images of spread fibres from each condition. Similar results were obtained in five independent experiments (cordycepin treatment—2). **c** Distribution of fork speed rates in INO80-proficient (siGFP) and INO80-deficient (siINO80). Data is from five independent experiments (in cordycepin-treated cells—2), at least 250 fibres were measured per condition in each experiment. ****$p$-value < 0.0001, (two- tailed unpaired Student's $t$-test). **d** Schematic representation of the experimental setup used. PC3 cells were co-transfected with esiRNAs against either GFP (siGFP) or INO80 (siINO80) along with either a control (CTRL) or RNase H1-overexpressing (RNAseH1) vector. Two days later RNAse H1 expression was induced by doxycycline for 24 h. Cells were labelled with CldU for 5 min followed by IdU pulse for 20 min and subjected to DNA fibre labelling analysis. **e** Representative images of spread fibres from each condition. Similar results were obtained in four independent experiments. **f** Distribution of fork speed rates (kilobase/min) in siGFP and INO80-deficient cells transfected with control or RNAse H1 overexpression plasmids. Data are from 4 independent experiments, at least 250 fibres were measured per condition in each experiment. ****$p$-value < 0.0001, (two-tailed unpaired Student's $t$-test). **g** Schematic representation of the experimental setup. Cells were co-transfected and induced as in **d** and prior to labelling were treated with 5 µM vorinostat for 6 h. **h** Representative images of spread fibres from each condition. Similar results were obtained in three independent experiments. **i** Distribution of fork rates from (**h**, at least 250 fibres were measured per condition in each experiment; ns non-significant, ****$p$-value < 0.0001, *$p$-value < 0.05, (two-tailed unpaired Student's $t$-test). **c**, **f**, **i** Kruskal–Wallis test $p$-value was < 0.0001. Data is presented as Tukey boxplot (box representing first quartile, median and third quartile, whiskers 1.5 times interquartile range).

damage and activate the S-phase checkpoint[25]. Chk1-Ser345, a downstream target of the checkpoint kinase ATR[26], was phosphorylated in siINO80 cells (Fig. 3a), suggesting increased DNA damage in unperturbed conditions. To test whether DNA damage during S-phase in the absence of INO80 is associated with R-loops, siControl and siINO80 cells were co-transfected with either the empty vector (EV) or the RNAse H1 o/e plasmid and grouped in replicating and non-replicating populations based on positive and negative EdU staining respectively (Fig. 3b). Quantitative immunostaining analysis was conducted against γH2A.X, a marker of DNA damage (Fig. 3c–e). Overexpression of RNAse H1 in control cells partially activated the S-phase checkpoint as expected[27], without affecting γH2A.X levels (Fig. 3a, c–e). Depletion of INO80 significantly increased γH2A.X signal in S-phase cells but not in non-S-phase cells(Fig. 3c–e). RNase H1 overexpression in siINO80 cells decreased the levels of Phospho-Chk1-S345 to levels comparable to control cells overexpressing RNAse H1 and significantly reduced the intensity of γH2A.X (Fig. 3a, c–e). This results indicate that R-loops induce DNA damage and activate the S-phase checkpoint in the absence of INO80.

We further tested whether DNA damage induced by replication stress in INO80-depleted cells[17] is dependent on R-loops. Control and siINO80 cells overexpressing RNAse H1 were treated with hydroxyurea (HU), a drug that depletes the dNTP pools, leading to replication stress and DNA damage[28,29]. Loss of INO80 led to an increase in γH2A.X intensity in HU-treated S-phase cells, but not in non-S-phase cells (Supplementary Fig. 5b–d). RNase H1 overexpression significantly reduced both the intensity of γH2A.X signal and the percentage of γH2A.X-positive cells in control and siINO80 cells (Supplementary Fig. 5b–e). Likewise, inhibition of transcription by α-amanitin resulted in a significant decrease in the intensity of γH2A.X inside S-phase in INO80-depleted cells either with or without HU (Supplementary Fig. 6). These results suggest that INO80 counteracts co-transcriptional R-loops to suppress replication-associated DNA damage in human cells.

**R-loops accumulate genome-wide in the absence of INO80.** To evaluate the levels of R-loops in the absence of INO80, immunofluorescence analysis was conducted in control and siINO80-treated PC3 cells using the S9.6 antibody (Fig. 4a, b). Consistent with other reports[30], S9.6 puncta were detected in the cytoplasm and the nucleus. The nuclear S9.6 signal was diminished upon RNase H1 overexpression (Supplementary Fig. 7a). Quantification of the nuclear S9.6 signal intensity revealed a significant increase in R-loops upon INO80 depletion (Fig. 4c, d). The

intensity of S9.6 signal was also increased in cells depleted for INO80 by viral shRNA compared to a non-targeting shScrambled Control (Supplementary Fig. 7b–d). In vitro treatment with recombinant RNAse H eliminated the increased S9.6 signal in siINO80 cells (Supplementary Fig. 7d). Pulse labelling of cells with the uridine analogue 5-ethynyluridine (EU), which is incorporated into newly synthesized RNA[31], revealed no significant changes in transcriptional activity between control and siINO80 cells (Supplementary Fig. 7e), indicating that the increase in R-loop abundance in the absence of INO80 is not due to elevated transcription rates.

We asked whether depletion of INO80 leads to an increase in R-loops inside S-phase. Analysis of R-loop intensity in EdU positive and negative cells showed enhanced accumulation of R-loops in siINO80 cells both outside and inside S phase (Fig. 4e). Therefore, R-loops that accumulate throughout the cell cycle in the absence of INO80 can be a potential source of genome instability during DNA replication.

To test whether INO80 prevents accumulation of R-loops formed at specific genomic loci, DNA:RNA immunoprecipitation (DRIP) assay was conducted in control and INO80-depleted cells by four independent lentiviral shRNAs (Fig. 4f, g). The promoter-proximal and termination regions of the *beta-actin* gene (*in1* and *pause* regions) and the *EGR1* gene are sites prone for R-loops formation[32,33]. Loss of INO80 induced a reproducible increase in R-loop enrichment at the *beta-actin in1* and *pause* regions, as well as in the *EGR1* gene (Fig. 4g). In contrast, no increase in R-loops was observed at the 5′ region upstream the *beta-actin* gene promoter in INO80-depleted cells when compared to control cells (Fig. 4g). The increase in DRIP-qPCR signal observed at the *beta-actin* and *EGR1* genes upon INO80 depletion was diminished upon treatment of the genomic DNA with recombinant RNAse H prior to DRIP (Supplementary Fig. 7f). These results suggest that INO80 counteracts accumulation of R-loops forming at R-loop prone sites.

**Nuclear colocalization of INO80 with R-loops.** We asked whether INO80 associates with nuclear R-loops. Immunofluorescence (IF) samples stained with the S9.6 antibody (R-loops) and anti-INO80 were imaged using STED nanoscopy and analysed for colocalization between INO80 and R-loops (Fig. 5a). The increased resolution of STED at ~50 nm in our conditions, compared to confocal imaging (~250 nm), allows discrimination between 'true' and 'false' colocalization events with high level of certainty. Colocalization between INO80 foci and R-loop foci by STED was readily observed (Fig. 5a), while multiple colocalization events between INO80 and R-loops visualised by confocal were

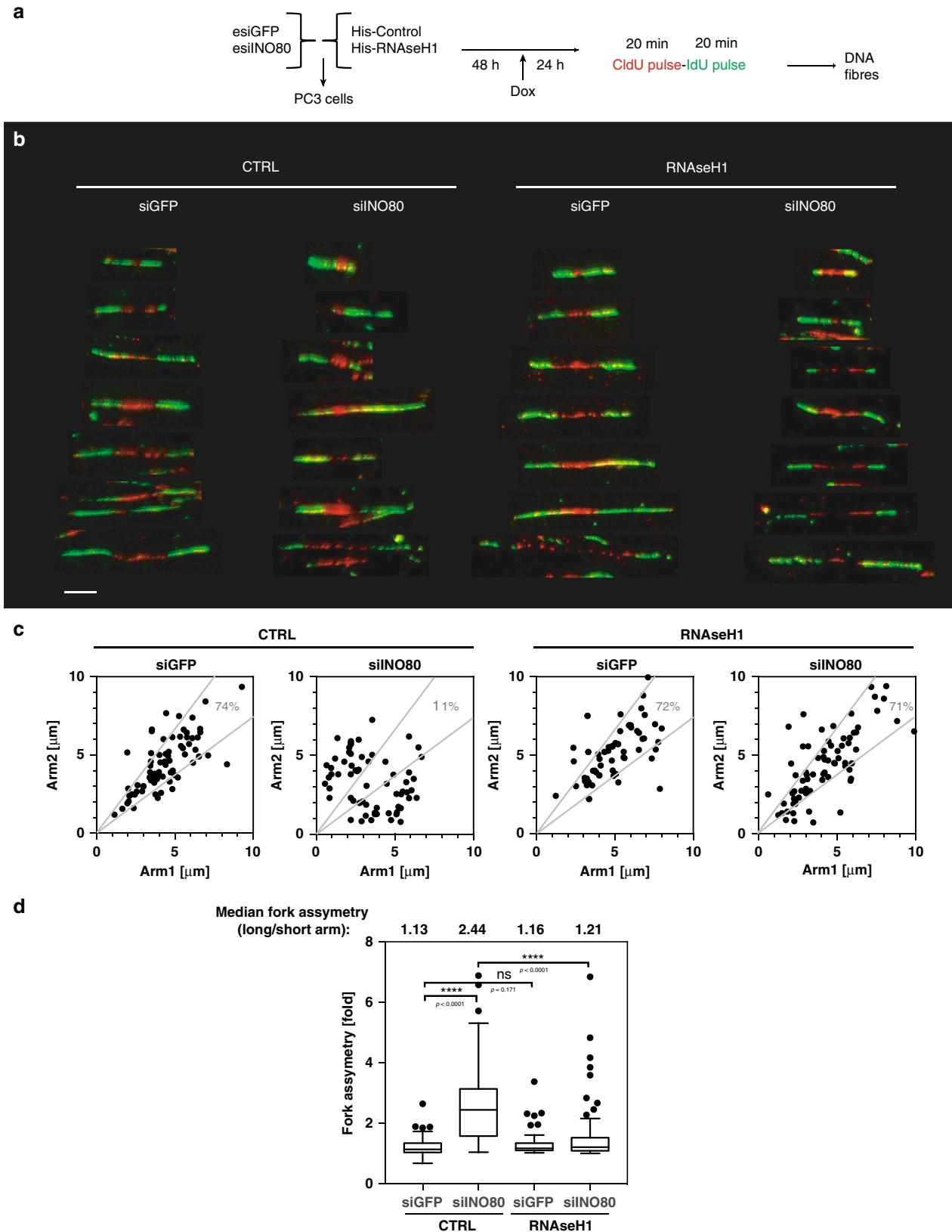

found to be separate, distinct foci when resolved by STED (Supplementary Fig. 8). To distinguish between random and non-random co-localization events, we conducted a Van Steensel's cross-correlation function analysis (CCF)[34]. Co-localization events between the STED imaged channel (INO80 or S9.6) and confocal imaged EdU were random, as expected (Supplementary Table 1).

Contrary, the global colocalization between the STED INO80 and STED S9.6 R-loop signals was not random (Supplementary Table 1). This suggests true R-loop:INO80 colocalization events.

Only a subset of INO80 foci co-localized with R-loops. To quantify the percentage of R-loop objects with overlapping volume with INO80 objects, we created 3D volume objects for

**Fig. 2 Fork asymmetry in Ino80-depleted cells depends on R-loops. a** Schematic representation of the experimental setup used. PC3 cells were co-transfected with esiRNAs against either GFP (siGFP) or INO80 (siINO80) along with either a control (CTRL) or RNAse H1-overexpressing (RNAse H1) vector. Two days later RNAse H1 expression was induced by doxycycline for 24 h. Cells were labelled with CldU for 20 min (red) followed by IdU pulse for 20 min and subjected to DNA fibre labelling analysis. **b** Representative pairs of sister replication forks were assembled from different fields of view and were arbitrarily centred on the position of origin. Scale bar 5 μm. Similar distribution of paired forks was observed in three independent experiments. **c** Scatter plots of the distances covered by right-moving and left-moving sister forks during the CldU pulse in Ino80-proficient or deficient cells expressing or not RNAse H1. The central areas delimited with grey lines contain sister forks with less than a 25% length difference. The percentage of symmetric forks is indicated. **d** Relative fork asymmetry. Fork asymmetry is expressed as the ratio of the longer arm to the shorter one for each pair of sister replication forks, ****$p$-value < 0.0001; n.s non-significant (two-tailed unpaired Student's $t$- test). Numbers above boxes indicate the median of the ratio of the longer to shorter arm. Data is presented as Tukey boxplot.

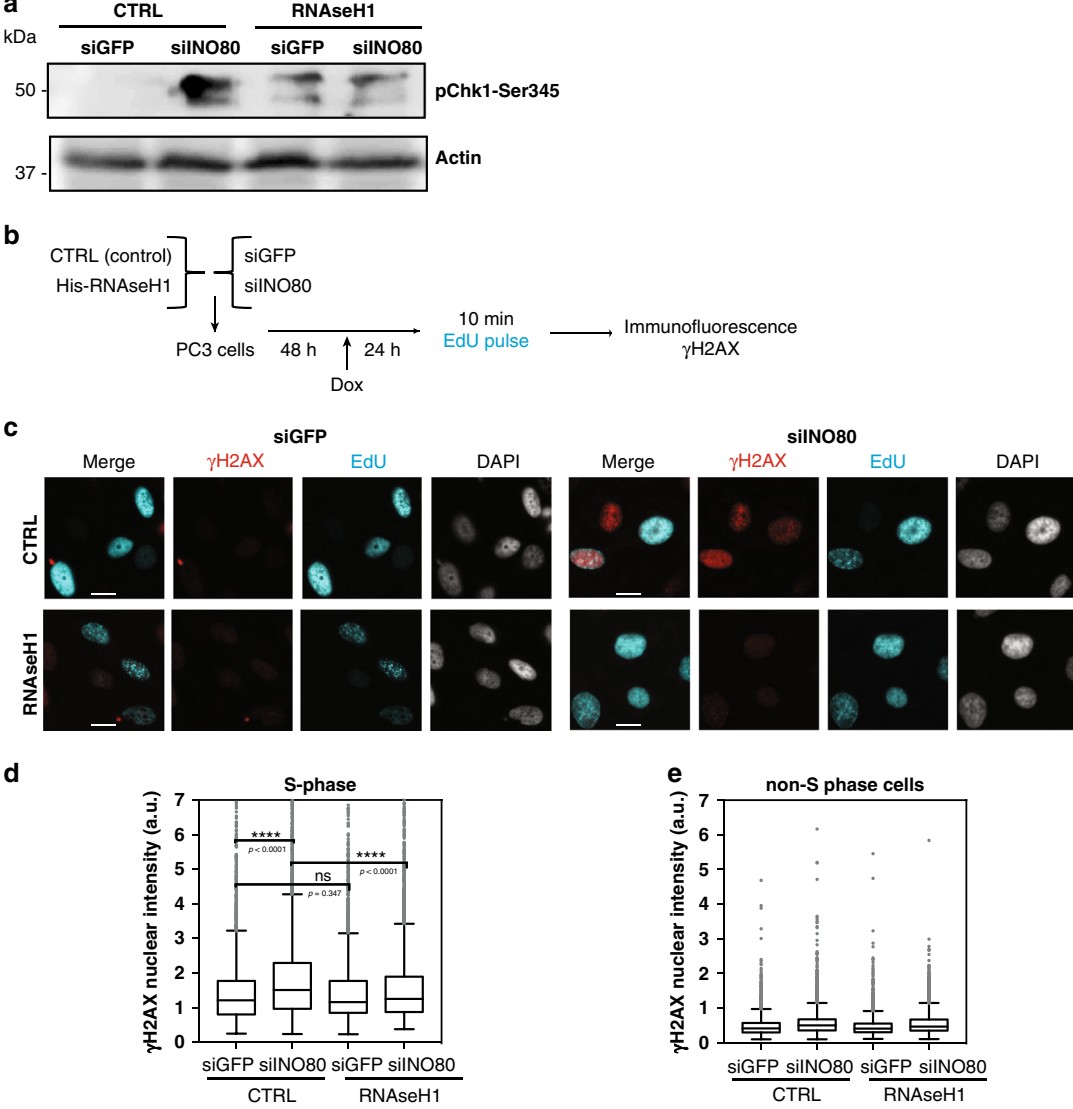

**Fig. 3 Replication stress-induced DNA damage in INO80-deficient cells is caused by R-loops. a** PC3 cells were transfected with control vector (CTRL) and either esiRNA against GFP (lane 1) or INO80 (lane 2) or transfected with RNAse H1-expressing plasmid (RNAseH1) and esiRNA against GFP (lane 3) or INO80 (lane 4). Forty-eight hours later cells were induced with doxycycline and 24 h later analyzed by Western with an antibody against pChk1-Ser345. Similar results were obtained in two independent experiments. **b** Schematic representation of the experimental setup used. PC3 cells were co-transfected with esiRNAs against either GFP (siGFP) or INO80 (siINO80) along with either a control (CTRL) or RNAse H1-overexpressing (RNAseH1) vector. Two days later RNase H1 overexpression was induced by doxycycline for 24 h. To distinguish cells in S-phase, cells were labelled with 25 μM EdU, fixed and stained with an antibody against γH2AX and "clicked" with Alexa Fluor 488 azide. **c** Representative images of cells as in **b**. Scale bar 10 μm **d** Distribution of nuclear γH2AX staining intensities in S-phase cells; ****$p$-value < 0.0001, ns nonsignificant (two-tailed unpaired Student's $t$-test). At least 500 cells were measured per condition in each of three independent experiments. Data is presented as Tukey boxplot. **e** Distribution of nuclear γH2AX staining intensities in non-S-phase cells. Data in **d**, **e** are from three independent experiments following normalization (to median intensity of entire population of siGFP/CTRL sample in each experiment). Tukey boxplot is used. Source data are provided as a Source data file.

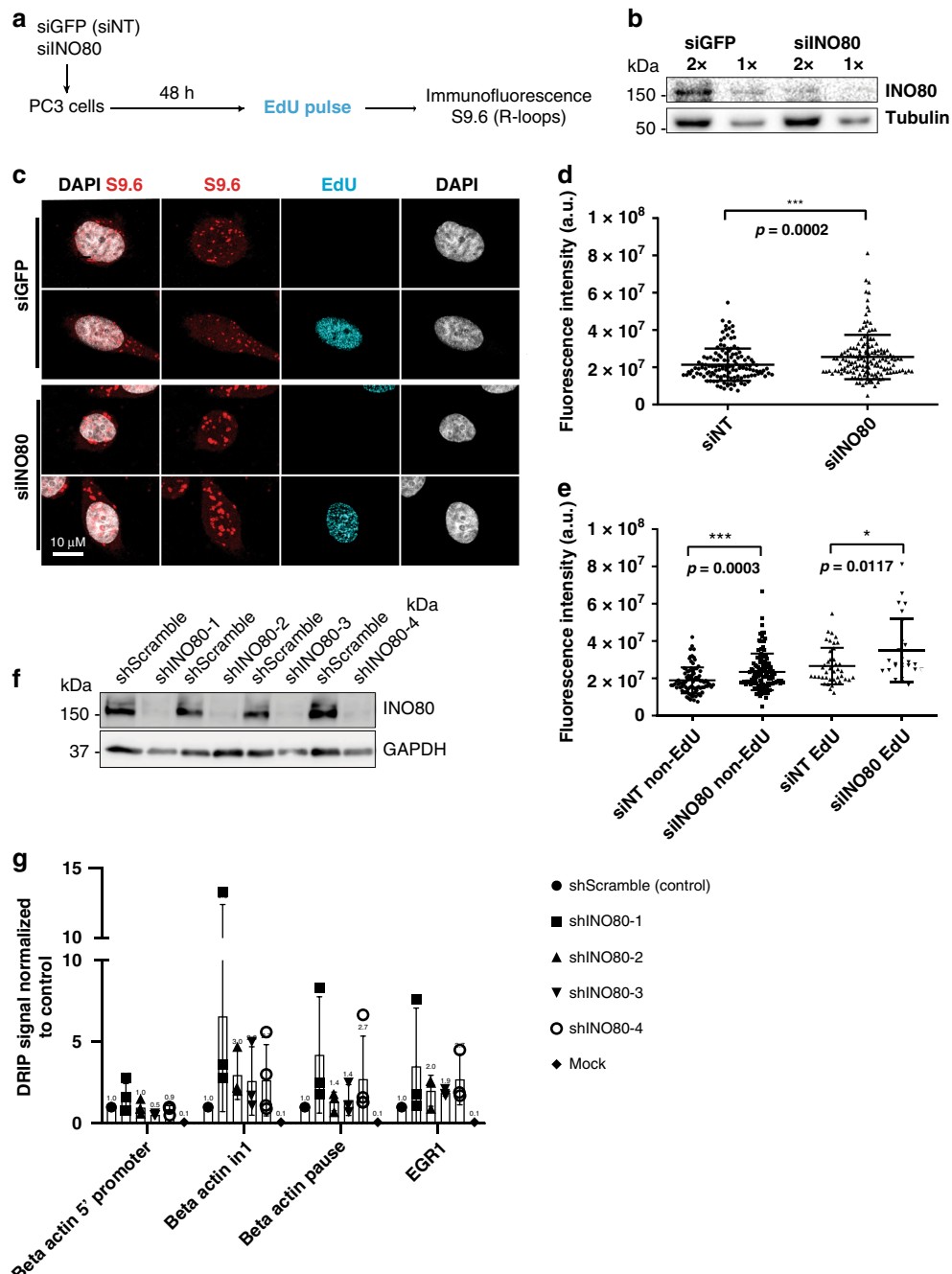

**Fig. 4 R-loops accumulate in INO80-depleted cells. a** Schematic representation of the experimental setup used. PC3 cells were co-transfected with esiRNAs against GFP (siNT) or INO80 (siINO80). Cells were pulsed with EdU for 15′ and immunofluorescence was carried out using the S9.6 antibody against R-loops. **b** Representative immunoblot of total extracts from PC3 cells transfected with esiRNAs against GFP (siGFP) or INO80 (siINO80) with an antibody against INO80 3 days after transfection. Similar results were obtained in five independent experiments. **c** Representative confocal deconvolved images EdU positive and EdU negative cells immunostained with S9.6 $n = 3$ biological replicates were quantified. Scale bar is 10 μM. **d** Total nuclear fluorescence intensity of R-loops in 3D volume, in control and INO80-depleted cells. DAPI was used as a mask to measure total fluorescence intensity of R-loops in 3D nuclear volume. Number of cells analyzed in each condition: siNT: $n = 133$ cells; siINO80: $n = 151$ cells. Data are from three independent experiments. $p$-value = 0.0002, (two-tailed unpaired Student's $t$-test). **e** Total nuclear fluorescence intensity of R-loops in 3D volume, in control (siNT) and INO80-depleted (siINO80) cells outside (non-EdU) and inside (EdU) S-phase. DAPI was used as in **d**. Number of cells analyzed in each condition: siNT (non-EdU): $n = 91$; siINO80(non-EdU): $n = 117$; siNT(EdU): $n = 42$; siINO80(EdU): $n = 26$. Data are from three independent experiments. \*\*\*$p$-values = 0.0003 \*$p$-value=0.0117, (two-tailed unpaired Student's $t$-test). **f** Representative immunoblot of PC3 cells transduced with control shRNA (ShScramble) RNA and four different shRNAs targeting INO80 (shINO80-1-4) following experimental setup shown in Supplementary Fig. 4b. Similar results were obtained in 4 independent experiments. **g** DRIP-qPCR was performed on the indicated shRNA-treated samples using the S9.6 antibody. Values for each region tested were normalized over shControl after correction for input DNA levels. Mock IP was conducted in shControl and shINO80 samples in the absence of S9.6 antibody. Data from three independent biological replicates are presented as mean values ± SD. Source data are provided as a Source data file.

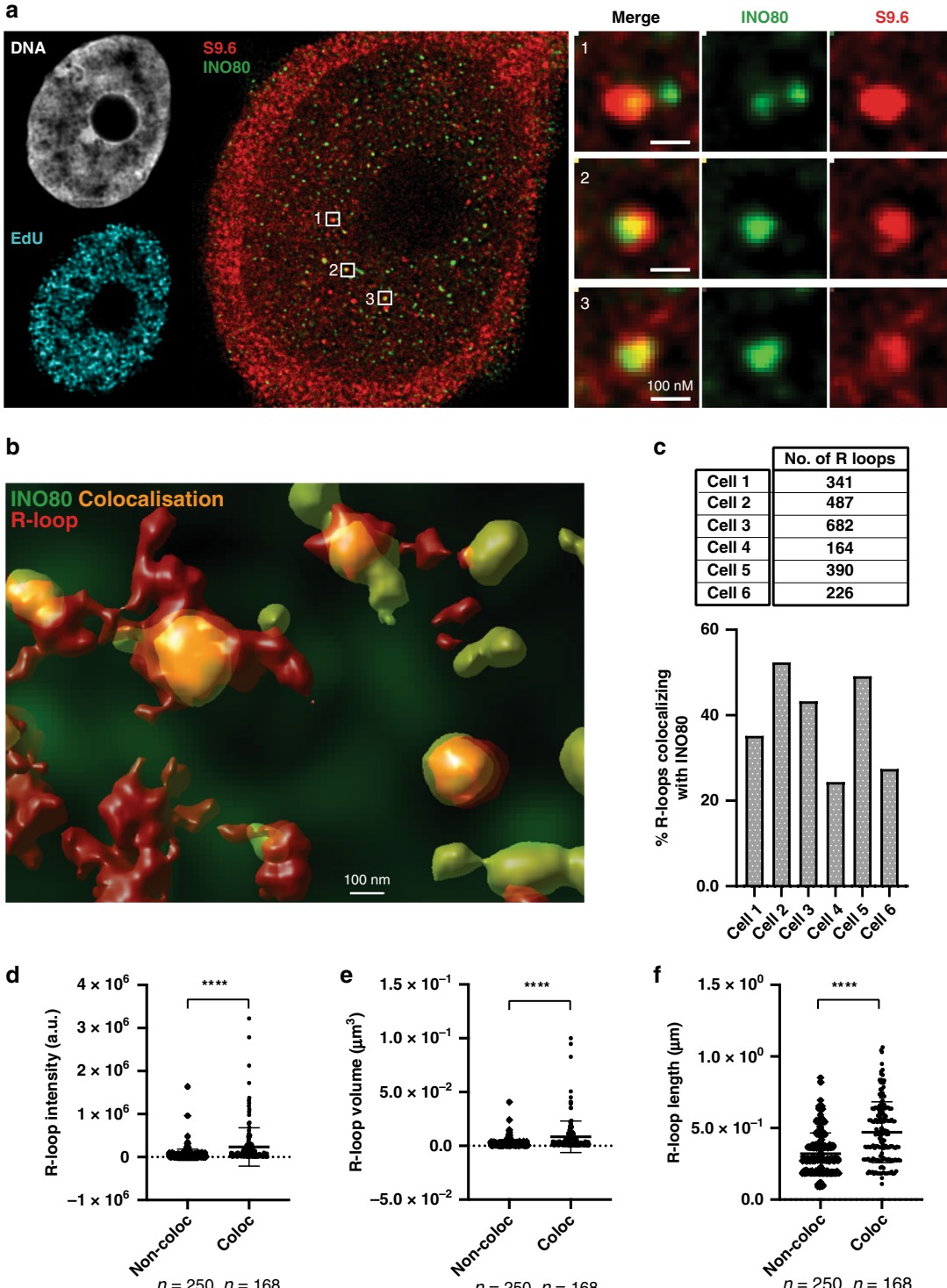

**Fig. 5 Colocalization of INO80 with R-loops by STED nanoscopy. a** Single image plane of confocal and STED imaging of PC3 cells. Cells were pulsed with EdU for 15' and immunostained with S9.6 and anti-INO80 antibodies and imaged using confocal (EdU) and STED (INO80 and S9.6). STED imaging pixel size is 11 nm in XY. Panel 1, 2 and 3 are magnified regions where INO80 and S9.6 co-localize. Scale bar = 100 nm $n = 6$ independent cells were analysed from three replicates **b** 3D volume surface model of INO80 and S9.6. Transparency of the INO80 volume (green) has been increased to enable visualization of overlapping regions (yellow). Similar results were obtained from six independent cells. **c** Upper panel. The number of R-loops per cell was determined in six independent EdU positive cells. Thresholds were generated based on fluorescence intensity within the Hoechst stained nuclear volume and excluded objects that were below 10 voxels in size as background. The number of R-loops (S9.6 volumes) which shared co-localizing voxel volumes with INO80 were quantified. Lower panel. Bar-graph indicating the percent of R-loops colocalizing with INO80 relative to the total amount of R-loops in each cell. **d–f** Comparative analysis of R-loop intensity sum **d**, voxel volume **e** and length **f** in colocalizing (coloc) or non-colocalizing (non-coloc) with INO80 in Cell 1 $n = 250$ non co-localising R-loops, $n = 168$ colocalising R-loops. (****adjusted $p$-value < 0.0001; one way ANOVA). Analysis of the cells 2–6 is in Supplementary Fig. 9. Data are presented as mean values ± Standard deviation (SD)".

R-loops and INO80 in the nucleus (Fig. 5b). The number of nuclear R-loops counted per cell in our analysis varied from 164-682 (Fig. 5c, upper panel). The percentage of R-loops which had some co-localizing volume with INO80 ranged between 25 and 52% (Fig. 5c, bottom panel), suggesting that a significant proportion of R-loops are bound by INO80.

To determine whether INO80 colocalizes with specific R-loops, we analysed the fluorescence intensity, volume and length properties of the 3D objects. All properties were significantly different between INO80 and R-loop objects (Supplementary Fig. 9a). For all cells analysed, the R-loops colocalized with INO80 were significantly more intense, had greater volume and greater length than their non-colocalizing counterparts (Fig. 5d–f and Supplementary Fig. 9b–f). These data suggest that the INO80 complex associates with the largest, most enriched, R-loop domains in the nucleus.

**Chromatin association of INO80 is promoted by R-loops**. To gain insight into the co-enrichment of INO80 and R-loops across the genome, we analysed the genome-wide association of INO80 and R-loops using published ChIP-seq data for INO80 and DRIP-seq data for R-loops in mouse embryonic stem cells (mESCs)[35,36]. Comparative analysis for INO80 and R-loop enrichment at protein-coding gene bodies revealed a significant positive correlation ($p = 1.1 \times 10^{-206}$, Fig. 6a), suggesting that INO80 is recruited to transcribed genes with high propensity to generate R-loops. Partial correlation analysis performed on the INO80 ChIP-seq and DRIP-seq data over RNA-seq data, indicated that the correlation between INO80 and R-loop enrichments remained highly significant when controlled for gene expression levels ($p = 5.5 \times 10^{-160}$, Supplementary Fig. 10a–c). Therefore, the positive correlation detected between INO80 and R-loop enrichment is not indirectly due to their mutual association with transcription. Visualisation of overlayed INO80 ChIP-seq and DRIP-seq reads confirmed that INO80 and R-loops are co-enriched at the beta-actin gene and other mRNA genes (Fig. 6b and Supplementary Fig. 10d).

To illuminate the genome-wide association of INO80 with R-loops, we analysed INO80 and R-loop enrichment across mESC chromatin states at 200 bp resolution. We segmented the genome into 20 chromatin states based on the combinations of 14 histone marks, 3 cytosine modifications and INO80 peaks[37] (Fig. 6c). The maximum enrichment of INO80 was found to be in states 10 and 15 (Fig. 6c). State 10 is characterised by high levels of histone marks such as H3K4me3 and H3K9ac which are associated with active promoters and characterise the transcription start site (TSS) of genes. State 15 is solely defined by INO80 enrichment (Fig. 6c). When the enrichment of R-loops with or without INO80 was analysed, R-loops were primarily enriched without INO80 in most of the chromatin states ("DRIPseq+INO80" and "DRIPseq" peaks respectively, Fig. 6d). However, the greatest association of R-loops with INO80 as well as the highest enrichment in R-loops were observed in state 15 (Fig. 6d). These results indicate that INO80 and R-loops strongly associate even outside annotated TSS and gene bodies and suggest that INO80 is recruited at genomic regions enriched in R-loops.

The Ruvbl1 and Ruvbl2 subunits of INO80 bind to RNA:DNA hybrid structures in vitro[38]. We, therefore, tested whether R-loops promote the binding of INO80 to chromatin. PC3 cells stably expressing either a control plasmid or the doxycycline-inducible RNAse H1 plasmid were subjected to differential salt fractionation after addition of doxycycline (Fig. 6e). Analysis of the different subcellular fractions showed that the amount of INO80 found in the high salt fraction, which represents soluble nuclear proteins and loosely-associated chromatin proteins, was increased

upon overexpression of RNAse H1 (Fig. 6f, g). Concurrently, the fraction of INO80 tightly bound to chromatin was significantly reduced by approximately two-fold (Fig. 6f, g). This suggests that R-loops promote stabilization of INO80 on chromatin.

**Artificial tethering of INO80 promotes R-loop resolution**. We hypothesized that INO80 may promote resolution of R-loops. Employing U2OS cells carrying the 256x-LacO tandem array[39], we devised an assay to monitor changes in R-loop enrichment upon artificial tethering of the INO80 complex. The LacO-LacI array has been reported to be a site of replication stress[40]. LacI-fused proteins bind LacO, while incorporating eGFP in the system enables visualisation of the LacO locus (Fig. 7a). S9.6 IF in LacO-U2OS cells expressing LacI-GFP demonstrated that the LacO array is enriched in R-loops (Fig. 7b). We next expressed LacI-eGFP tagged versions of RNAse H1 or the INO80E subunit of the INO80 complex in LacO-U2OS cells and the S9.6 signal overlapping with the eGFP-LacI signal was quantified. Tethering of LacI-eGFP-RNAse H1 to the LacO array led to a significant decrease in the intensity of the underlying R-loop signal (Fig. 7b, c). This indicates that R-loops at the LacO site can be suppressed by artificial recruitment of factors promoting their resolution. Tethering of LacI-eGFP-INO80E reduced the R-loop signal to levels similar to LacI-eGFP-RNAse H1 (Fig. 7b, c). This suggests that INO80 is directly involved in the downregulation of R-loops. Depletion of INO80 in LacO-U2OS cells expressing LacI-eGFP-INO80E resulted in increased enrichment of R-loops at the LacO site, suggesting that intact INO80 complex is required for the suppression of R-loops (Supplementary Fig. 11).

To understand how INO80 regulates R-loops, we monitored the dynamics of R-loops at the lacO locus in live cells. LacO-U2OS cells were transfected with a plasmid expressing the RNA Binding Domain of RNAse H1 fused to DsRed (RBD-DsRed) (Fig. 7d). The RBD construct allows monitoring R-loop enrichment[41] in live cells. In agreement with our S9.6 IF results, the RBD-DsRed signal also accumulated at the LacO locus (Fig. 7e). We co-transfected cells with RBD-DsRed and either LacI-eGFP or LacI-eGFP-INO80E and performed time-lapse imaging every 6 min for 25 h, 24 h after transfection (Fig. 7f; Supplementary movies 1 and 2). The RBD-DsRed intensities relative to the colocalizing eGFP signal were quantified throughout the time-course of the experiment in single cells. Changes in R-loop signal intensity were analysed by calculating the Fold-Change in the relative DsRed fluorescence Intensity (FC-I) between every time point and its previous one (FC-I = $I^{t2}/I^{t1}$) and plotted in log2 scale as FC-I(log2) (Fig. 7g). The fold-change in R-loop intensity indicates the number of R-loops created minus the R-loops resolved during the specific time period. A positive FC-I(log2) value ($I^{t2} > I^{t1}$) suggests net formation of new R-loops across the LacO site between the two time points. A negative FC-I (log2) value ($I^{t2} < I^{t1}$) indicates that resolution of R-loops is greater than formation for the specific time period. In both LacI-eGFP and LacI-INO80 the mean FC-I(log2) values were positive, indicating that R-loops are constantly formed at the LacO site. However, the mean FC-I(log2) value in cells transfected with LacI-INO80E was significantly smaller than the mean FC-I(log2) value for LacI-eGFP (Fig. 7g, Total), indicating INO80 actively counteracts R-loop formation. Tethered INO80 could affect R-loop dynamics through either suppressing their formation, or promoting their resolution. To distinguish between these possibilities, positive and negative FC-I(log2) values were clustered separately in LacI-eGFP and LacI-INO80E cells. The positive FC-I(log2) values in the LacI-INO80E cells were significantly smaller than in the LacI-eGFP (Fig. 7g). If INO80 affects R-loop formation but not R-loop resolution, then the same

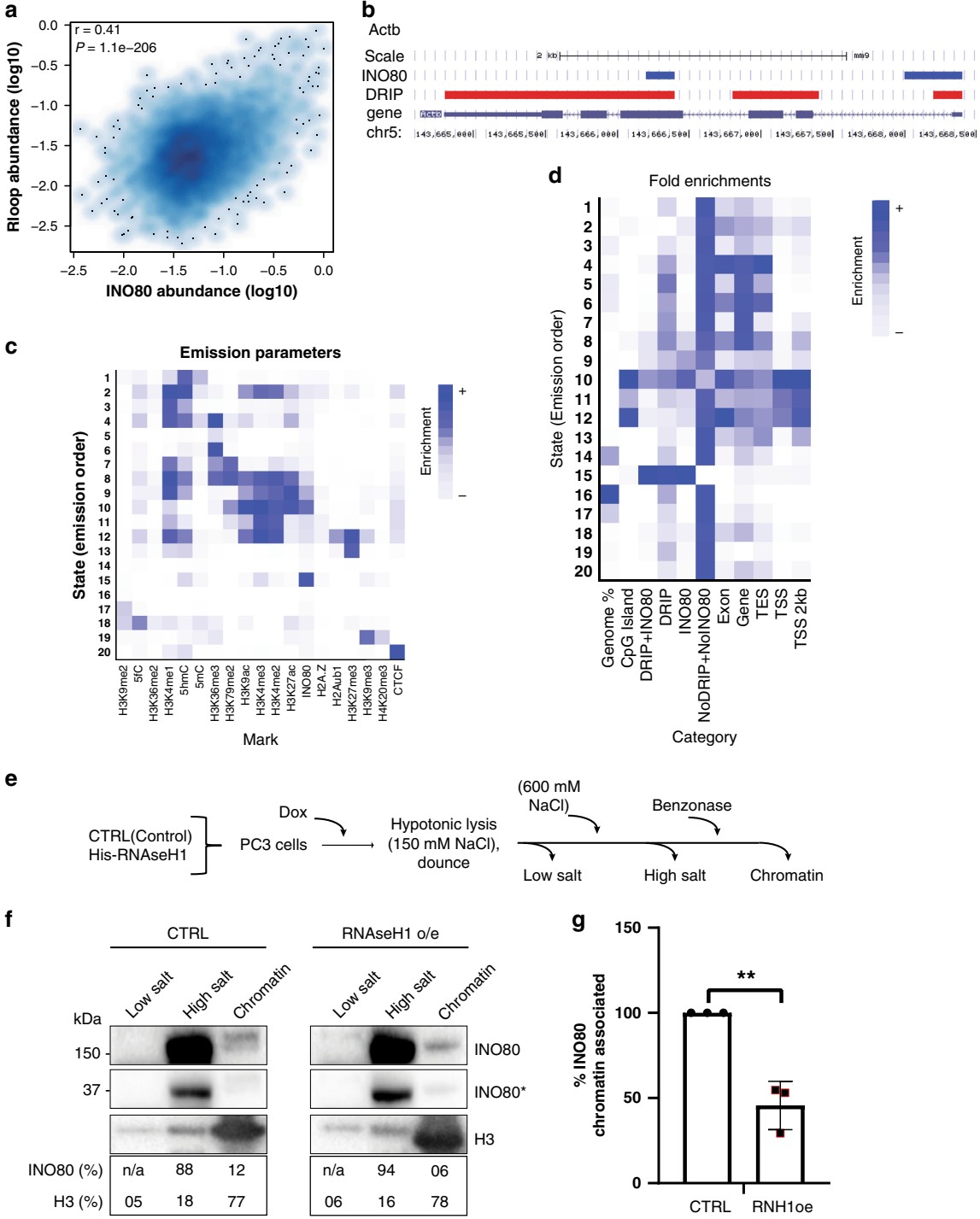

‘amount’ of resolution should be applied to both the LacI-eGFP and LacI-INO80E datasets. In that case, because LacI-INO80E cells have a lower mean positive FC-I(log2) value than LacI-eGFP, the negative FC-I(log2) values upon tethering INO80 should be greater. However, when the negative FC-I(log2) values were analysed, no significant difference was found between LacI-INO80E and LacI-eGFP (Fig. 7g). This contests the hypothesis that resolution of R-loops is independent of INO80, and suggests that tethering INO80 does not impact R-loop synthesis. Thus, the dynamics of the R-loop signal at the LacO locus observed upon artificial recruitment of INO80 suggest a role for INO80 in promoting turnover of R-loops.

**R-loop resolution by INO80 promotes cancer cell proliferation.** We asked whether the role of INO80 in cancer cell growth[14] is associated with removal of R-loops. Proliferation was monitored in cells that were either co-transfected with the RNAse H1 overexpression plasmid or not and depleted for INO80. siRNase H1 cells were also analysed to assess the effect of R-loop accumulation in cancer cell proliferation (Supplementary Fig. 12a). In addition, we calculated the expected change in growth predicted in the case that the two factors are not functionally associated.

Silencing of either INO80 or RNase H1 did not affect the proliferation of human embryonic kidney HEK293 cells, however

**Fig. 6 R-loops associate with INO80 genome-wide and promote INO80 binding to chromatin. a** Smoothed scatterplot showing the pairwise correlation between DRIP-seq (R-loops) and INO80 ChIP-seq abundances at gene bodies in mESCs. **b** Genomic enrichment profile of INO80 and R-loops (DRIP) signals across the beta-actin (*Actb*) gene in mouse ESCs. **c** Heat map for model parameters of ChromHMM, indicating the relative emission probability of each mark/feature to each state. **d** Genome-wide enrichment of DRIP-seq (R-loops) and INO80 ChIP-seq peaks at 200 bp binning resolution across different chromatin states based on marks shown in Fig. 4c. Columns indicate the relative percentage of the genome represented by each chromatin state and relative fold enrichment for different types of annotation. INO80 + DRIP: overlapping INO80 and DRIP-seq peaks; DRIP: DRIPseq peaks not overlapping with INO80 peaks; INO80: INO80 peaks not overlapping with DRIPseq peaks; NoDRIP+NoINO80: regions of the genome lacking both DRIPseq and INO80 peaks; TES Transcription End Site; TSS Transcription Start Site. **e** Schematic representation of the differential salt fractionation setup used. PC3 cells stably expressing either control vector or a Tet-inducible RNase H1-expressing vector were treated by doxycycline for 24 h. Sequential subcellular fractionated lysates were isolated in low salt (150 mM), high salt (600 mM) and from benzonase-digested pellet to release the tightly bound chromatin-associated factors. Fractions analyzed by immunoblot for INO80 and histone H3. **f** Immunoblot of subcellular fractions from control and RNHaseH1 expressing cells from a representative chromatin fractionation experiment. Histone H3 was used as a marker for chromatin enrichment. INO80 and H3 were analyzed from the same gel. INO80* indicates lighter exposure of the INO80 immunoblot. *N* = 3 biological replicates were performed with similar results. **g** Bar graph indicating the change in INO80 enrichment in the chromatin fraction between control and RNAse H1 overexpressing cells. The amount of INO80 detected in the chromatin fraction of the control cells was set arbitrarily to 100%. Data are from three independent experiments. **p-value = 0.0027 two-tailed unpaired Student's *t*-test). Data are presented as mean values ± SD. Quantification by ImageJ. Source data are provided as a Source data file.

both depletions compromised the growth of PC3 cells (Fig. 8a, b and Supplementary Fig. 12b, c). RNAse H1 overexpression rescued the growth of siINO80 PC3 cells by almost four-fold compared to the expected growth value (Fig. 8b), indicating that the proliferation defect caused by INO80 depletion is rescued by removal of R-loops.

INO80 has been reported to promote growth of NRAS oncogene mutant-driven melanoma cells[21]. Depletion of INO80 in the NRAS mutant WM1361 melanoma cell line compromised cellular growth, while RNAse H1 overexpression in the siINO80 WM1361 cells rescued growth by approximately three-fold (Fig. 8c). R-loop-induced replication stress is a reported hallmark of E2 estrogen-positive MCF7 cancer cells[7]. Disruption of INO80 led to a severe growth defect in MCF7 cells (Fig. 8d). Notably, overexpression of RNAse H1 in INO80-depleted MCF7 cells strongly rescued their proliferation defect (Fig. 8d). These results suggest that INO80-dependent resolution of R-loops is required for proliferation of cancers characterized by dysregulated transcription.

Although proliferation of PC3 cells is severely compromised and DNA damage accumulates upon depletion of INO80 or RNase H1[42], loss of either factor led to a minor increase in cell lethality (Fig. 8e and Supplementary Fig. 12d). We therefore hypothesized that R-loop-induced DNA damage is efficiently repaired in cancer cells, thus averting cell death. DNA damage associated with R-loops is repaired by the base excision repair (BER) pathway[43], which requires the AP endonuclease APE1/yAPN1 and the homologous recombination repair factor Rad52[44]. We therefore tested whether combined inhibition of APE1 and Rad52 sensitizes cancer cells lacking INO80 or RNase H1 to death. PC3 and HEK293 cells depleted of either INO80 or RNase H1 were treated with the APE1 and Rad52 inhibitor 6-hydroxy-DL-Dopa[45,46] (DL-Dopa) and assessed for lethality. None of the treatment combinations induced increased cell death in HEK293 cells (Supplementary Fig. 12e). A 15-20-fold increase in lethality was observed in siINO80 and siRNAse H1 PC3 cells treated with DL-Dopa (Fig. 8e and Supplementary Fig. 12f). In contrast, inhibition of the DNA damage checkpoint factor ATR which also safeguards against genotoxic R-loops[47], did not induce further cell death in siINO80 PC3 cells (Supplementary Fig. 12g). This synthetic lethality phenotype suggests that proliferating cancer cells with unresolved R-loops rely on the BER pathway for their viability. Taken together, our results suggest that R-loop resolution facilitated by INO80 ameliorates DNA damage at sites of transcription-replication conflicts to promote cancer cell proliferation and prevent cell death.

## Discussion

Here, we elucidate a role of the human INO80 complex in DNA replication. Our study reveals that by counteracting accumulation of R-loops, INO80 prevents genotoxic conflicts between transcription and the replication machinery and promotes efficient DNA synthesis. Our study supports the idea that INO80 defines a pathway for the removal of R-loop structures from chromatin that is critical for maintenance of genome integrity and cancer cell proliferation.

A recent study using an in vitro eukaryotic DNA replication system on a nucleosomal template has suggested that INO80 promotes replisome progression through chromatin in the absence of transcription[48]. While this possibility cannot be ruled out by our study, our in vivo evidence suggests that INO80 facilitates DNA replication by averting collisions between the fork and co-transcriptional R-loops: Firstly, RNAse H1 overexpression rescued DNA replication progression and suppressed fork stalling in INO80-depleted cells (Figs. 1 and 2). Secondly, the replication-associated DNA damage of INO80-depleted cells was significantly relieved by overexpression of RNAse H1 or chemical inhibition of transcription (Fig. 3). These results support a model where, by counteracting R-loops, INO80 removes a critical barrier to DNA replication and suppresses replication-associated DNA damage, rather than facilitating the repair of DNA damage.

While impaired fork progression in siINO80 cells is rescued by overexpression of RNAse H1 in normal conditions (Fig. 1), we do not anticipate that every fork encounters R-loops in the absence of INO80, as the increase in fork asymmetry also indicates (Fig. 2). Loss of INO80 leads to constitutive activation of the ATR/Chk1 DNA synthesis checkpoint pathway in normal conditions (Fig. 3), which reduces global DNA synthesis rates[49] and slows down elongation even at unchallenged forks[50]. It is thus likely that stalling of forks following collisions with R-loops in the absence of INO80 induces activation of the ATR/Chk1 pathway, triggering an overall slowdown of replication fork movement. In addition, our observation that R-loops contribute to the high levels of DNA damage observed in HU conditions, in both normal and INO80-depleted cells (Supplementary Fig. 5), makes it plausible that INO80 is not indiscriminately targeting R-loops, but it is specifically required to suppress those R-loops that can potentially interfere with forks in a genotoxic, head-on orientation[5,51].

Several data indicate a direct involvement of INO80 in downregulation of pre-existing R-loops. INO80 reduces the enrichment of R-loops formed at specific genomic sites, such as the beta-actin gene (Fig. 4). STED nanoscopy revealed that

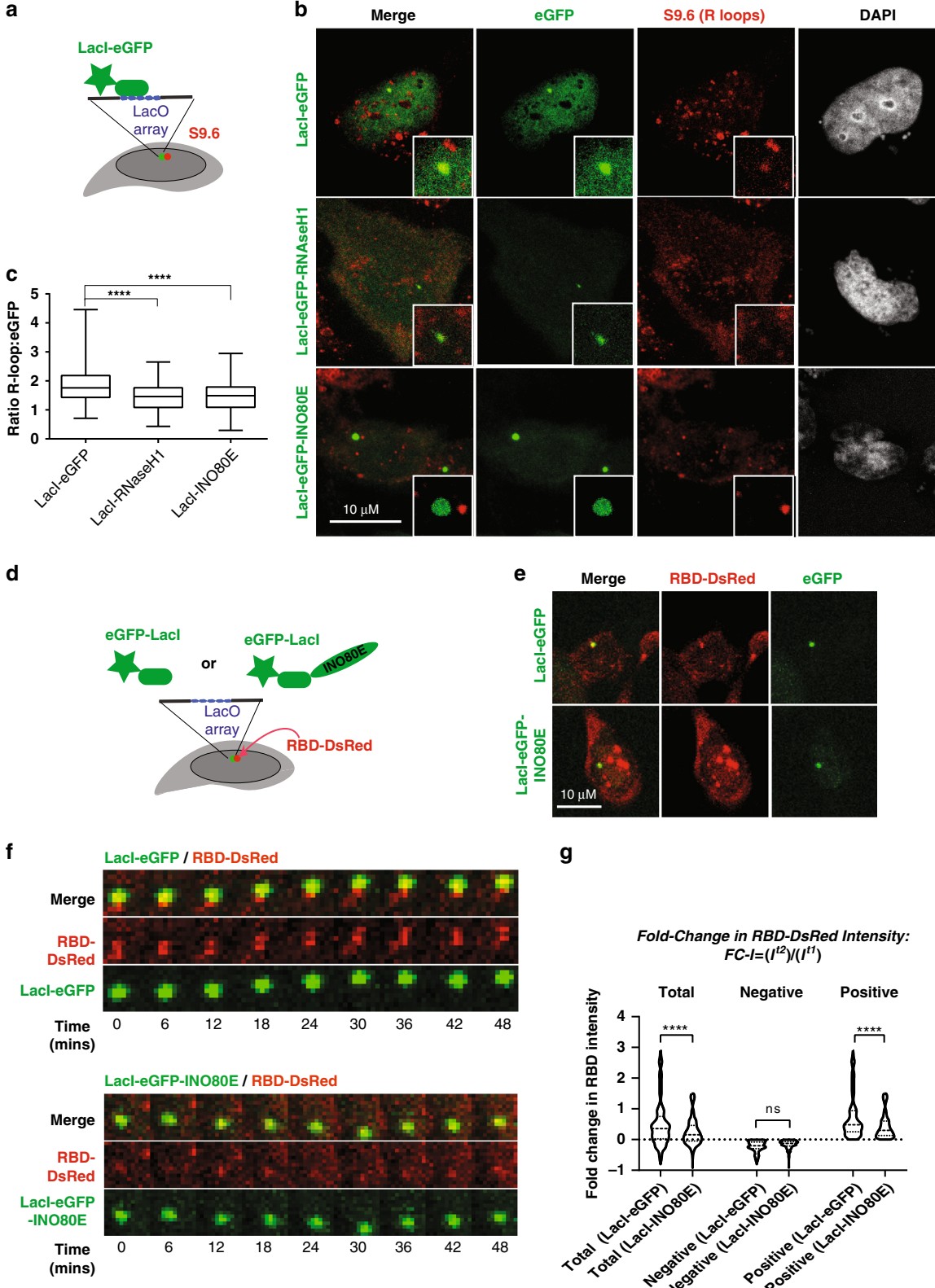

INO80 binds up to 50% of the total R-loop sites detected at ~50 nm resolution, while 3D analysis of the STED data suggests a preferential colocalization of INO80 with the largest and most enriched R-loop sites (Fig. 5). Genomic analysis at 200 bp resolution in mESCs indicated enrichment of INO80 at specific R-loop-enriched genomic regions, including the beta-actin gene, while RNAse H1 overexpression compromised the association of

INO80 with chromatin (Fig. 6). Taken together, our single cell, biochemistry and genomic analyses suggest the presence of a regulatory mechanism for recruitment of INO80 by R-loops across the genome.

Unexpectedly, our genomics analysis revealed an uncharacterized chromatin state that is defined by INO80 and is highly enriched in R-loops but is not associated with neither the R-loop-

**Fig. 7 Dynamic R-loop turnover at the LacO locus by INO80 tethering. a** Schematic representation of the experimental assay. LacI-eGFP tagged INO80E or RNHseH1 were transfected into U2OS cells harbouring a 256xLacO array. Immunostaining with S9.6 after 24 h allows visualization of R-Loops at the LacO locus. **b** Representative immunostained images of LacO cells transfected with eGFP-LacI, eGFP-LacI- INO80E, eGFP-LacI-RNHaseH1. Scale bar is 10 μm. N = 4 biological replicates were performed with similar results **c** Boxplot showing the S9.6 signal intensity relative to the underlying eGFP-LacI-tagged proteins intensity at the LacO locus in the respective conditions, as described in **a**. Plot shown is min to max values with line at median. Number of cells per condition shown: eGFP = 97; RNase H1 = 116; INO80E = 100. N = 3 experiments ****adjusted *p*-value < 0.0001; one way ANOVA test. **d** Schematic representation of the experimental assay. LacO-carrying U2OS cells were co-transfected with RBD-DsRed and either LacI-eGFP or LacI-eGFP-INO80E plasmids and live cell imaging carried out in Z stacks imaged every 6 min. **e** Representative live images of LacO cells transfected with LacI-eGFP or LacI-eGFP-INO80E *n* = 3 biological replicates gave similar results. Scale bar is 10 μm. **f** Images of selected time-points are shown from a representative live imaging experiment. Merge shows LacO co-localisation with LacI-eGFP or LacI-INO80E cell during 48 min of the time course analysis for RBD-DsRed and LacI-tagged proteins. Merged images of whole cells are in Supplementary Figure 11d. Montage is a zoom of the LacO site from the representative cell in **e**. **g** Upper panel: formula for calculating Fold-Change in Intensity (FC-I). Intensity change of the RBD-DsRed signal normalized over the underlying eGFP signal at the locus was measured at 6-minute intervals throughout the course of the experiment for a total of 1500 min. FC-I was calculated as the normalized RBD-DsRed signal at each timepoint relative to the previous timepoint. Lower panel: Total, negative and positive FC-I(log2) values for RBD-DsRed signal in LacI-eGFP and LacI-INO80E cells. The total number of FC-I(log2) values was approximately 900 for both LacI-eGFP and LacI-INO80E. Positive FC-I(log2) values are 70–75% of the total. Negative FC-I(log2) are 25-30% of the total. N = 5 independent live cells quantified for each condition. ****adjusted *p*-value < 0.0001; ANOVA with Kruskal–Wallis test.

enriched 5′ and 3′ ends of gene bodies[52], nor with the gene body itself (Fig. 6). Characterisation of this chromatin state is an important step towards elucidating the cellular mechanisms controlling R-loop metabolism and promoting genomic stability.

Our observation that artificial tethering of INO80 at the lacO array led to reduced enrichment of R-loops in *cis* suggests that INO80 binds to genomic regions enriched for R-loops in order to promote their removal (Fig. 7). Time-lapse analysis of R-loops at the lacO site suggested that onsite recruitment of INO80 did not suppress formation of R-loops but instead triggered their turn-over. Although we cannot formally exclude that binding of the RBD-DsRed construct at the lacO array is compromised in the presence of LacI-INO80E, the results obtained from our kinetics analysis (Fig. 7g) argue against this possibility. If RBD-dsRd binding was adversely affected by LacI-INO80E, and the resolution kinetics remained the same upon binding of either LacI-GFP or LacI-INO80E, we would expect to see an increase in negative values of RBD intensity change in the LacI-INO80E compared to lacI-GFP. However, we observe similar negative values in the LacI-GFP and LacI-INO80E cells (Fig. 7g). Moreover, the well-documented role for mammalian INO80 in transcriptional activation[21,35], makes it unlikely that INO80 decreases the abundance of R-loops at the lacO site by repressing transcription. The INO80 complex has been reported to physically interact with RNA:DNA helicases such as DDX5 or DDX59[53–55]. Given that INO80 promotes extraction of ubiquitinated RNA Polymerase II from chromatin[16], it is plausible that INO80 coordinates resolution of R-loops with removal of stalled RNA Polymerase II.

Human INO80 has been linked to opening up chromatin structure[14]. Evidence suggests that the chromatin surrounding R-loops adopts a compacted nucleosomal structure[56,57]. We observed that chemically induced decompaction of chromatin by SAHA/Vorinostat rescued the DNA replication defect of INO80-depleted cells in an epistatic manner with RNase H1 over-expression (Fig. 1). This supports the possibility that the chromatin remodelling activity of INO80 facilitates decompaction of the repressive chromatin landscape at R-loop enriched sites, revealing an intriguing aspect of R-loop regulation by chromatin.

Recent reports have shown that oncogenic and hormone-dependent transcription, in HRAS overexpressing cells and in breast cancer MCF7 cells respectively, leads to enhanced forma-tion of R-loops and increased R-loop-dependent DNA damage during DNA replication[6,7]. The fact that these cancer cells are able to sustain sufficient DNA synthesis rates for their pro-liferation under such highly genotoxic conditions suggests that cancer cells have established mechanisms to cope with the increased occurrence of genotoxic transcription-replication con-flicts. Our findings that (i) INO80 counteracts genotoxic R-loops to promotes proliferation of prostate, breast and melanoma cancer cells (Fig. 8), and (ii) INO80 depletion is synthetically lethal with Rad52/APE1 inhibition suggest a chromatin-based R-loop resolution mechanism in cancer cells that suppresses their inherent predisposition for DNA damage during S-phase. Whe-ther resolution of R-loops by INO80 regulates oncogenic tran-scription and enables coordination of dysregulated gene expression with DNA replication in cancer cells is an exciting possibility.

In conclusion, by identifying INO80 as a molecular link between cellular proliferation and silencing of R-loops, our study provides insight into how cancer cells balance transcription with replication, enabling unlimited growth in the presence of inherent replication stress conditions.

## Methods

**Cell culture, transfections and treatments**. Human PC3 cells were grown in DMEM (Gibco) supplemented with L-glutamine, 10% fetal bovine serum, 1 mM sodium pyruvate and antibiotics in 5% CO$_2$ atmosphere at 37 °C. Cells were treated with 0.5 mM hydroxyurea to induce replication stress. To inhibit transcription α-amanitin (2 μg/ml) and cordycepin (50 μM) were used. Induction of the Tet-ON promoter was achieved with 1 μg/ml doxycycline for 24 h. 5-Ethynyl-2′-deoxyur-idine (EdU)to label S-phase cells was used at 25 μM final concentration and 5-ethynyl uridine (EU) was used at 1 mM to assess the overall nascent transcription.

EsiRNAs targeting the coding regions of human INO80 (3440-3894, transcript NM_017553.1) or EGFP (132-591) were synthesized following standard procedures[58,59]. Primers used to amplify the targeted regions were selected using Riddle database[60]:

hIno80:
5′-TCACTATAGGGAGAGTGTGGAGCATCAGACCTCAG;
5′-CACTATAGGGAGACCCTGCTTTGTCTGCCCTAAG

hIno80–3′UTR:
5′-TCACTATAGGGAGAGAAGTGGAAATGTCCAGCAGGG; 5′-TCACTAT
AGGGAGACCTGGGAACACAACTGCCTGTGG;

hArp8: 5′-TCACTATAGGGAGAGGGCACGCTCCTACAATAAGC; 5′-TCA
CTATAGGGAGACGTGCTGCTTAAGCCACTTCC.

GFP:5′-TCACTATAGGGAGAGGCCTGAAGTTCATCTGCACCA; 5′-TCACT
ATAGGGAGAC TGCTCAGGTAGTGGTTGTCG

The siRNAs targeting RNAse H1 were as follows:
RNase H1 (#1): 5′-CUGUCUUGCUGCCUGUACU-3′
RNase H1 (#2): 5′-GAAGUUUGCCACAGAGGAU-3′

INO80 MISSION shRNA shRNA pLKO1 Plasmid DNA targeting INO80 were purchased from Merck. The sequences used were:

sh1: CCGGGCAGTTGTGTTCCCAGCAATTCTCGAGAATTGCTGGGAAC
ACAACTGCTTTTTG

sh2: CCGGGCCCAGAAGAACTGTAAGGAACTCGAGTTCCTTACAGTTC
TTCTGGGCTTTTTG

sh3: CCGGGCTGCTATATCAGGCACTAAACTCGAGTTTAGTGCCTGATA
TAGCAGCTTTTTG

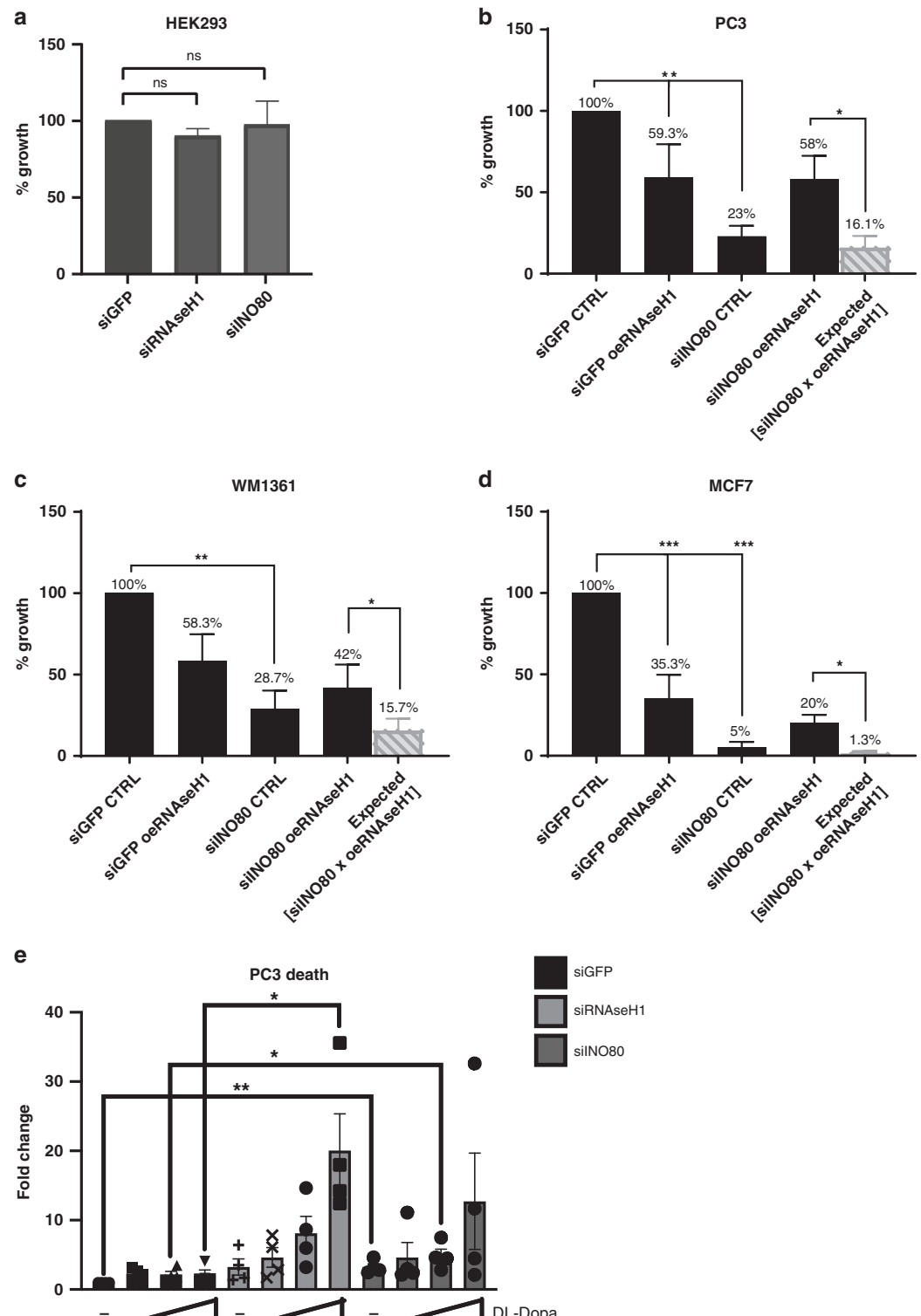

sh4: CCGGCGTAACCTGTTTCTCACCAATCTCGAGATTGGTGAGAAAC
AGGTTACGTTTTTG

Lentiviral particles were produced following standard procedure[61]. PC3 cells were transduced with viral supernatant and grown under 2ug/ml puromycin selection for 8 days prior to harvesting for western blot and DRIP experiments.

The inducible expression of RNAse H1 has been achieved using pEBTet-BLAST-RNAse H1-myc/His and pEBTet-EGFP-BLAST was used as control.

Antibodies against RNAse H1 (Invitrogen PA5-30974) and GAPDH (Santa Cruz Biotechnology) were used.

In the rescue experiment INO80 was expressed from pCMV-3XFLAG-hINO80 —a gift from Joan Conaway (Addgene plasmid # 44149)[62]. As empty plasmid pcDNA3.1 was used.

Quantities of Lipofectamine and esiRNAs for efficient knockdown were optimized using esiRNA against Eg5 (Kif11). Typically, 60 pmol of esiRNA and 2 µl of Lipofectamine 2000 were used per well in a 24 well plate (500 µl transfection volume). Knockdown of INO80 was assessed by Western blotting. Co-transfections of esiRNA and plasmids were carried out as above with 45 pmol esiRNA and 200 ng plasmid DNA.

**Antibodies.** anti-INO80, Proteintech 18810-1-AP (used at 1:1000 dilution)
anti-INO80, Abcam ab118787 (used at 1:500 dilution)
S9.6 custom made from S9.6 hybridoma, ATCC® HB-8730
anti-RNAse H1, Invitrogen PA5-30974 (used at 1:1000 dilution)

**Fig. 8 RNAse H1 overexpression rescues cancer cell proliferation in the absence of INO80. a–d** Cells from the indicated lines were co-transfected with control (siGFP) or INO80 targeting (siINO80) esiRNAs along with either a control (CTRL) or RNAse H1-overexpressing (oeRNAse H1) vector. Proliferation was monitored by cell counting 96 h after transfection. Expected values to test for independent effect of INO80 depletion and overexpression of RNAse H1 in cell growth are calculated according to[70], as: V = [%growth siINO80] × [%growth oeRNAse H1]. Data are from 3 independent experiments. Exact $p$ values: Fig. 8a = 0.1 for siRNaseH1 and 0.99 for siINO80 vs siGFP, Fig. 8b $p$-value = siGFP vs siINO80 = 0.047, p-value siGFP vs siINO80 = 0.007; $p$-value siINO80-siRNaseH1 vs predicted = 0.05. Figure 8c: p-value siGFP vs siINO80 = 0.009, siINO80-siRNaseH1 vs predicted = 0.046 Fig. 8d: p-value siGFP vs siRNaseH1 = 0.0005, p-value siGFP vs siINO80 > 0.0001, siINO80-siRNaseH1 vs predicted 0.0245 *p value < 0.05; **p-value < 0.01, ***p-value < 0.005; two-tailed unpaired Student's $t$-test. Data are presented as mean values ± SEM **e** Cell death analysis. PC3 cells transfected with control (siGFP), RNase H1-targeting (siRNaseH1) or INO80-targeting (siINO80) siRNAs for 24 h were treated with increasing concentrations of DL-Dopa for further 7 days incubation and analysed for cell growth (Supplementary Fig. 9f) and cytotoxicity. Cell death was calculated for cytotoxicity fluorescence values normalized to the respective relative cell growth. Fold change cell death values were calculated by setting untreated control cells arbitrarily to 1. Concentrations used for DL-Dopa inhibitor: non-treated (−), 1 μM, 2 μM and 5 μM. Data are presented as mean values ± SD, measure of centre is mean. Data are from three independent experiments. $P$ values calculated by unpaired two-tailed $t$-test. (siGFP v siINO80 $P$ value **$p$ = 0.0036, siGFP 2 μM v siINO80 2 μM $P$ value *$p$ = 0.0464, siGFP 5 μM v siRNaseH1 5 μM $P$ value *$p$ = 0.0156.

anti-GAPDH, Santa Cruz Biotechnology 6C5 (used at 1:2000 dilution)

anti-Tubulin, Santa Cruz sc-5286 (used at 1:5000 dilution)

anti-Histone H3, Merck Millipore 06-755 (used at 1:5000 dilution)

anti-phosphoH2AX (Ser 139), clone 2F3, BioLegend, 613401, lots: B219075, B219074

anti-BrdU, clone B44, Becton Dickinson, 347580, lots: 9172603, 7157935 (used at 1:25 dilution)

anti-BrdU, Abcam, ab6326, lot: GR3289293-3 (used at 1: 400 dilution)

anti-actin, Abcam, ab8226, lot: GR3299142-1 (used at 1: 2500 dilution)

anti-ssDNA, Millipore MAB3034, lot: 3209139 (used at 1: 100 dilution)

anti-pChk1(S345), Cell Signaling 2341T, lot: 8 (used at 1:1000 dilution)

anti Myc-tag, clone 9B11, Cell Signaling 2276S, lot: 24 (used at 1:1000 dilution)

anti-mouse IgG DyLight488, Abcam, #ab96879, lot: GR252791-1 (used at 1:200 dilution)

anti-rat IgG DyLight594, Abcam #ab96889, lot: GR263830-2 (used at 1:400 dilution)

anti-rabbit IgG DyLight594, Abcam #ab96873

anti-mouse IgG AlexaFluor647, Molecular probes #A-31571 (used at 1:200 dilution)

AffiniPure Donkey Anti-mouse IgG AlexaFluor647, Invitrogen A-31571

AffiniPure Donkey Anti-Rabbit IgG Alexa Fluor 594 Invitrogen A-32740

anti-rabbit IRDye680, Li-Cor,#926-32221 (used at 1:10000 dilution)

anti-mouse IRDye800CW, Li-Cor, #926-32210 (used at 1:10000 dilution)

**DNA fibre labelling**. DNA fibre analyses were performed following standard protocol[63] with slight modifications. Briefly, exponentially growing PC3 cells were first incubated with 25 μM chlorodeoxyuridine (CldU) and then with 250 μM iododeoxyuridine (IdU) for the indicated times. Spreads were prepared from 2500 cells, suspended in phosphate buffered saline (PBS) at 1 × 10^6 cells/ml. Cell lysis was carried out in fibre lysis solution (50 mM EDTA and 0.5% SDS in 200 mM Tris-HCl, pH 7.5). DNA fibres were spread by tilting the slides ~25°until the drop of the fibre solution reached the bottom of the slide and let to dry. Dried slides were either stored at 4 °C or processed immediately. Slides were suspended in 2.5 M HCl for 80 min, washed in PBS, and then incubated in blocking buffer (5% bovine serum albumin in PBS) for 40 min. Primary antibodies—mouse anti-BrdU antibody (Becton Dickinson, cat # 347580) to detect IdU and rat anti-BrdU antibody (Abcam cat# Ab6326) to detect CldU—were diluted in blocking buffer and applied overnight. Slides were washed several times in PBS, incubated with secondary antibodies for 60 min. For fork asymmetry experiments fibre integrity was assessed by staining with an anti-ssDNA mouse antibody (Millipore, MAB 3034) followed by an anti-mouse Alexa fluor 647 secondary. Slides were mounted with ProLong Gold anti-fade reagent (Molecular Probes). Images were acquired with Axiovert 200 M microscope (Carl Zeiss) equipped with Axiocam MR3 camera (Carl Zeiss) or Dragonfly 500 microscope with iXon Camera (Andor). Fibre length measurements were carried out using AxioVision software (Carl Zeiss) or Image J. Speed conversion was carried out using conversion factor of 2.59 kb per μm. The formula used was speed = (length × 2.59)/time according to ref. [64].

**EdU and EU staining**. For EdU staining cells were labelled with 25 μM of the compound as indicated. The click reaction was carried out after immunostaining in a reaction mixture containing 100 mM Tris-HCl, pH 7.6, 4 mM CuSO_4, 10 μM Alexa Fluor 488 azide and 100 mM sodium ascorbate for 30 min at room temperature. Slides were then washed 3 times using 5% FBS in PBS to remove unbound reporter. To label newly synthesized RNA, cells were labelled with 1 mM EU for 3 h, fixed, permeabilized and stained in the click reaction mixture as above. Nuclei were counterstained with 0.5 μg/ml 4′,6-diamidino-2-phenylindole (DAPI) in PBS.

**Flow cytometry**. To analyse cell cycle profiles, cells were harvested by trypsinization and fixed in 70% ethanol. Before analysis cells were re-suspended in PBS, treated with RNAse A (20 μg/ml) and stained with propidium iodide (20 μg/ml). Data acquired using FACScalibur apparatus (Becton Dickinson). Analysis was done using FlowJo software, fitting Dean-Jett-Fox model[65] to data to determine the percentage of cells in each phase.

**LacI-LacO system**. Cloning was performed using the Gateway Cloning system (Thermo Fisher). INO80E (HsCD00352991) and RNAse H1 (HsCD00022287) Gateway DONR plasmids were purchased from the DNASU Plasmid Repository (https://dnasu.org/DNASU/Home.do). INO80E and RNAse H1 were subsequently recombined into a Gateway cloning adapted plasmid, pEGFP-LacI-GW which was created from pEGFP-LacI-NLS-VP16; a gift from Karsten Rippe (Addgene plasmid # 103836; http://n2t.net/addgene:103836; RRID:Addgene_103836).

The U2OS-lacO-ISceI-Tet19 (U2OS LacO) cell line was a generous gift from Dr Evi Soutoglou, IGBMC and described in[39].

The RBD-DsRed plasmid was constructed by PCR cloning the HB domain of RNAse H1 into the pDsRed-Express-C1 vector (Clontech) using the following primers:[41]

RNH1_HBF (5′-ACTCA GATCTGGGATGTTCTATGCCGTGAGG-3′)

RNH1_HBR (5′-ATTGAG TCGACGCTTGCTGATTTCCTGAC-3′)

**Immunofluorescence and image analyses**. The S9.6 antibody was purified from the S9.6 mouse hybridoma cell line (ATCC® HB-8730™), at the Protein Expression and Purification Core Facility in Institut Curie, France using the ATCC recommended growth conditions.

For immunofluorescence, the protocol from ref. [57] was used with the following modifications. Cells were grown on coverslips for 24–72 h. Cells were fixed using ice-cold MeOH and stored overnight at −20 °C. Coverslips were then washed 1x in PBS 1′, followed by 10′ wash in 50 mM NH_4Cl in PBS for 10′ at room temperature. Cells were washed 2x in PBS, then 1x in PBS-Triton X 100 0.1% for 5′ each. Cells were blocked for 30′ in 3% BSA in 0.1% PBS-TX. S9.6 antibody was used at 1:1000, anti-INO80 antibody (Abcam ab118787) was used at 1:1000 both diluted in 1% BSA in 0.1% PBS-TX. Secondary antibodies for confocal microscopy were from Jackson Immunoresearch Alexa Fluor® 647 AffiniPure Donkey Anti-mouse IgG (1:500-1500); Alexa Fluor® 594 AffiniPure Donkey Anti-Rabbit IgG (for INO80 confocal and STED) or for S9.6 STED, ATTO 647N (STED/GSD) Goat anti-mouse IgG (Active Motif Catalog No. 15038) at 1:100. DNA was labelled using Hoescht. Coverslips were mounted using Prolong Diamond (Thermo Fisher) for confocal imaging or Mowiol 4.88 (Calbiochem) mounting media was prepared and used for STED. Mowiol mounted slides were allowed to cure for 48 h prior to imaging. Using this protocol, the non-specific S9.6 signal was depleted from the nucleolus. Images were deconvolved prior to quantification. Deconvolution was performed with Huygens 18.04 from SVI (www.svi.nl). Quantifications of both total nuclear R-loops and EdU associated R-loops were carried out using 3D volumes with IMARIS image analysis software. Maximum intensity stack projections were used for presentation in Fig. 2.

Recombinant RNAse H (NEB #M0297) was used following standard protocol[66] with the modification that fixed cells on coverslips were digested with RNase H in RNAse H buffer for 2 h at 37 °C, after which coverslips were washed 3 ×10′ in PBS prior to immunofluorescence staining. Control samples were mock-treated with RNase H buffer.

To stain for γH2AX, cells were grown on coverslips, washed in PBS, fixed with ice-cold methanol for 7 min at −20 °C, permeabilized with 0.5% Triton X-100 in PBS for 5 min, washed with PBS and blocked in 5% bovine serum albumin (BSA) in PBS containing 0.05% Tween (PBS-T) for 1 h. Staining was done using mouse anti-γH2AX Ab (BioLegend) diluted 1:200 using the same dilution overnight at 4 °C. Slides were then washed 3 ×5 min in PBS-T and secondary IgG DyLight 594 were used at 1:500 dilution for 1 h at room temperature.

**Confocal Microscopy and STED Nanoscopy**. Super-resolution methodology was used for the visualization of R-loops and INO80. STED overcomes the diffraction limit of conventional confocal microscopy (*). This yields resolutions of ≥200 nm for visible light in the lateral dimensions (in x–y) and ≥500 nm, in the axial direction (in Z) (65). STED resolution is typically approximately 50 nm in XY and 150 nm in Z. Images were acquired on a Leica TCS SP8 STED 3X point scanning confocal nanoscopy with while light super continuum lasers and three STED depletion lasers (592, 660 and 775 nm) using STED WHITE HC PL APO CS2 100×/1.40 OIL lens. The DAPI and AF488 channels where acquired in confocal mode while the AF594 and ATTO647 channels where acquired in confocal and STED mode. Colocalization analysis were performed following the protocol in[67]. Images were deconvolved prior to quantification. Deconvolution, colocalization and particle analysis was performed with Huygens 18.04 from SVI (www.svi.nl).

*Abbe, E. "Beiträge zur Theorie des Mikroskops und der mikroskopischen Wahrnehmung" [Contributions to the Theory of the Microscope and of Microscopic Perception]. Archiv für Mikroskopische Anatomie (in German). Bonn, Germany: Verlag von Max Cohen & Sohn. 9 (1): 413–468.

**Cell proliferation**. For proliferation analysis cells were seeded in 96-well plates 2000 cells/well. IncuCyte measurements of cellular occupation of the wells were taken every 3–6 h. Cell growth rate was normalised to the time point zero and additionally in a separate set of experiments cell numbers were counted at 96 h to assess cellular proliferation.

**Cytotoxicity assay and genotoxic agents**. After removal of 96-well plates from the IncuCyte, cytotoxicity was evaluated using CellTox Green® cytotoxicity assay (Promega) following the manufacturer's guidelines. CellTox green dye was diluted to 1:2000 in assay buffer and 25 μL added to each well of cells followed by incubation for 30 min at 37 °C under a humidified atmosphere with 5% $CO_2$. Fluorescence was measured at Ex:485, Em:520 nm in POLARstar® Omega microplate reader (BMG LABTECH).

The following genotoxic agents were used in the assays: 6-Hydroxy-DL-DOPA (TOCRIS), ATR inhibitor:(R)−4-(2-(1H-indol-4-yl)-6-(1-(methylsulfonyl) cyclopropyl)pyrimidin-4-yl)-3-methylmorpholine (MedKoo). Transfected cells were incubated for 24 h before the addition of genotoxic agents for further 7 days incubation at 37 °C under a humidified atmosphere with 5% $CO_2$.

**Statistical tests**. GraphPad PRISM version 7 and 8 were used to calculate significance and prepare graphs for presentation in figures.

**DRIP-qPCR**. DRIP assay was performed by performing IP with the S9.6 antibody in 2.5 μg genomic DNA and following the protocol of experiment 5 in ref. [66]. The primers used in the qPCR assay are as follows:

EGR1 (F): CATAGGGAAGCCCCTCTTTC
EGR1 (R): CTTGTGGTGAGGGGTCACTT
beta-actin 5′prom (F): CCA CCT GGG TAC ACA CAG TCT
beta-actin 5′prom (R):TGT CCT TGT CAC CCT TTC TTG
beta-actin in1 (F): CGG GGT CTT TGT CTG AGC
beta-actin in1 (R): CAG TTA GCG CCC AAA GGA C
beta-actin pause (F): GGG ACT ATT TGG GGG TGT CT
beta-actin pause (R): TCC CAT AGG TGA AGG CAA AG

**ChIP-seq, DRIP-seq and RNAseq analyses**. INO80 ChIPseq data retrieved from ArrayExpress E-GEOD-49137 dataset[35]. Fastq files from input control (GSM1194195: SRR942473-SRR942474) and INO80 pull down (GSM1194194: SRR942470- SRR942472) were aligned and processed according to ref. [37], with 200 bp peaks called with ChromHMM binarization function (see https://github.com/EpiStemNet for details). DRIP-seq data was downloaded from GEO GSE67581 dataset[36] and it was processed using the same pipeline. In this case, we used GSM1650022 (SRR1952485) sample as ChIP-seq "pull down" and the RNaseH-treated sample (GSM1650023:SRR1952486) as the equivalent of the "input". The presence or absence of each feature (INO80 or R-loops) per gene was obtained counting the number of 200 bp peaks intersecting with RefSeq genes using BED-Tools v2.25.0[68] and Bioconductor's package *TxDb.Mmusculus.UCSC.mm9.knownGene* (version 3.2.2). The ChromHMM model was generated from ref. [37] including INO80 peaks as an additional feature. Normalised single cell RNAseq for mESCs was downloaded from Espresso database (https://www.ebi.ac.uk/teichmann-srv/espresso)[69] and we used the median value across all cells.

**Reporting summary**. Further information on research design is available in the Nature Research Reporting Summary linked to this article.

## Data availability

The authors declare that the data supporting the findings of this study are available within the paper and its Supplementary information files. All datasets generated during and/or analysed during the current study are available from the corresponding author on reasonable request.

Espresso Database:https://www.ebi.ac.uk/teichmann-srv/espresso

INO80 and DRIP ChIPseq data URLs: https://www.ebi.ac.uk/arrayexpress/experiments/E-GEOD-49137/ https://www.ncbi.nlm.nih.gov/geo/query/acc.cgi?acc=GSE67581 Chromatin State data URL: http://epistemnet.bioinfo.cnio.es/download/Source data are provided with this paper.

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

## Acknowledgements
We thank Neil Perkins and members of the Papamichos-Chronakis lab for constructive discussions. We thank Claus M. Azzalin for the RNAse H1-His-Myc overexpression plasmid. We thank Evi Soutoglou for the LacOU2OS cell line. Work in MPC lab is supported by Liverpool University, Newcastle University and a Wellcome Institutional Support Fund Award. Work done in A.G.'s lab was supported by Bulgarian National Science Fund grant # DN11/17. Work in the PP lab is supported by grants from the Agence Nationale pour la Recherche (ANR), the Ligue Contre le Cancer (équipe label-lisée), SIRIC Montpellier Cancer (INCa Inserm DGOS 12553) and the MSDAvenir fund. ULM was funded by the Royal Society RG170342. IHL was funded from the European Union's Horizon 2020 research and innovation programme under the Marie Skłodowska-Curie grant agreement No. 754510. Work in the DR lab is supported by a Wellcome Trust Seed Award in Science (206103/Z/17/Z). Work of JMGH was funded by a Wellcome Trust Investigator Award [106951/Z/15/Z] and a Royal Society Wolfson Research Merit Award.

## Author contributions
L.P., A.G. and M.P.C. developed the project, designed and interpreted the experiments, and co-wrote the manuscript. L.P. conducted experiments for Figs. 4–6 with the help of S.G. A.G. conducted experiments for Figs. 1–3 with the help of RH. RB-P assisted with imaging and conducted analysis for Fig. 5. D.R. and I.H.M. conducted analysis for Fig. 6. P.P. and J.M.H. provided material. U.L.M. and K.V. conducted experiments for Fig. 8. All authors provided input in the manuscript preparation.

## Competing interests
The authors declare no competing interests.

**Additional information**

