## [Peer Review File · Nature Communications]

Reviewers' comments:

Reviewer #1 (Remarks to the Author):

This study reports that INO80 complex is involved in removing R-loops to prevent replication stress in human cells. RNAi depletion of INO80 subunit of this complex leads to increase in signal and to markers of replication stress such as replication fork slowing and gamma-H2AX. Ectopic overexpression of RNase H1 can rescue these phenotypes, as well as some of the lethal effect of depleting INO80 in cancer cell lines. There is some evidence of genome-wide co-localisation of R-loops and INO80, and that INO80 suppresses R-loop formation in cis.

On the positive side, this topic is timely and interesting. Firstly, knowledge on INO80 function in replication stress is limited and new insights will greatly contribute to advancing the field. Secondly, this study also uses some potentially interesting new approaches for R-loop detection.

On the negative side, the work as it stands seems quite superficial. The experiments lack controls, leading to not very convincing data. Furthermore, the effect of INO80 depletion on slowing down of replication fork speeds and cell death is much more pronounced than the effect of INO80 depletion on R-loop signal. This suggests that there are other, more important, mechanisms besides R-loops at play. There clearly appears to be some interplay between INO80 and RNaseH1, but mechanistically, the effects could be quite indirect, for example as a result of more DNA damage or stress generally in the absence of INO80. The data as presented cannot clarify that question. Furthermore, none of the other subunits of the INO80 complex have been depleted to see whether this causes similar phenotypes. The data would be much more convincing with such a broader approach. Finally, the study is using cancer cell lines, but that is not the same as directly investigating whether INO80 is involved in response to oncogenes. Therefore, the impact of the findings is not as high as it might be suggested in the introduction.

Specific points:

- There are no controls for off-target effects on the INO80 siRNA. It is reported that the esiRNA approach used here is more specific and with fewer off-target effects, but how sure can we be of this? It would be much better to use two different siRNA sequences to target INO80 or to rescue effects by INO80 re-expression.
- Figure 4: The effects of INO80 depletion on R-loop signals are quite moderate and these experiments are lacking controls. Specifically, it would be good practice to use treatment with recombinant RNaseH as controls for the S9.6 quantification in Fig. 1b.
- Fig. S4b: The S9.6 slot blot experiment needs controls such as DNA loading control and RNaseH treatment as well. Ideally, other methods of R-loop quantification should also be used, such as DRIP qPCR with the appropriate controls, to further strengthen the argument.
- Figure 5: The high-resolution microscopy data very interesting, but results are only shown for 4 cells. Is this staining and co-localisation really that clear in the majority of cells and are the data representative? These results would also need more verification by other methods. If INO80 co-localises with R-loops, this should also show up in standard DRIP qPCR experiments. For example, it would be worth using ChIP and DRIP to test for INO80 enrichment at known R-loop loci such as across the beta-Actin gene (K. Skourti-Stathaki Mol Cell 2011) and at some of the loci that have been identified to contain R-loops and INO80 in the analysis in Figure 6C.
- Figure 7: The imaging data using LacI tethering are very interesting, but is it possible that the larger fusions with RNH1 or INO80 sterically prevent binding of the RBD-dsRed? It looks as if RBD-dsRed may be not binding properly in presence of the longer construct. This assay would need a control experiment with a protein fusion that cannot remove R-loops, such as catalytically inactive RNaseH1 mutant.
- Figure 7f: What are the two different "merge" panels?
- The discussion states that "chemical inhibition of transcription rescued almost entirely the defects in DNA replication progression and in fork symmetry observed in INO80-depleted cells during normal S phase (Fig. 2 and 3)." But the effect of chemical transcription inhibitors in Fig. 2f is actually quite small.

Reviewer #2 (Remarks to the Author):

In this manuscript by Prendergast and colleagues, "Resolution of R-loops by INO80 promotes DNA replication and maintains cancer cell proliferation and viability", the authors set out to address how cancer cells manage to resolve R-loops that form obstacle to replication fork progression and further show that the suppression of r-loops by INO80 maintains proliferation of cancer cells. To this end, the authors mainly used RNaseH over-expression system for rescue experiments to show that R-loops accumulation is the main source of replication-associated DNA damage in Ino80 depleted cancer cells. Further, they performed DNA fibre analysis to observe replication fork progression defects in silno80 cells. From this they conclude that accumulation of R-loops is the main cause of replication defects, which also leads to fork asymmetry and both of which can be rescued by either over-expression of RNaseH1 enzyme or by using HDAC inhibitors or by using global transcription inhibitors. Additionally, they generated lacO-lacI system to study the R-loop resolution at lacO site. They generate fusion proteins of Ino80E subunit with LacI for artificial tethering of Ino80 complex to the lacO site and generate truncated RNA binding domain of RNaseH1 (RBD-DsRed) to signify R-loops in live-imaging assays. Using this system, they claim to show that artificial tethering of Ino80 helps in efficiently resolving R-loops rather than preventing them from forming at first place. Furthermore, the authors show that depletion of Ino80 results in synthetic lethality in multiple cancer cell lines, which is due to accumulation of unresolved R-loops and can be rescued by RNaseH1 over-expression.

Although the premise of work is interesting and some of the findings being novel however some key conclusions have been based on data which is not very convincing. For example- it is conceptually hard to accept the fact that the transcription inhibitors and HDAC inhibitors that enhances transcription in general, are rescuing the same replication defects shown for silno80 cells. Whereas HDAC inhibitors have been suggested to accumulate R-loops and slow down replication fork progression, previously (see more details below). Furthermore, the claim that Ino80 resolves R-loops in replicating cells mainly, is not strongly supported by their data. Other than this, there are frequent discrepancies in their own data within or in between figures, which has not been addressed by the authors. Overall, the work is interesting but requires addressing key issues with appropriate experiments with validations, to be considered for publication.

Major comments:

1. In Figure 1, authors claim that HU-mediated replication stress causes DNA damage, shown by γ -H2AX intensity and % of cells, in replicating (S phase) cells and not in outside of S phase cells upon knockdown of Ino80. This they claim is due to accumulation of R-loops as this was rescued by over-expression of RNaseH1 which resulted in significantly reduced, both the intensity and the percentage of γ H2AX positive cells (Fig. 1c-e). However, the percentage of S phase cells in both control and silno80 cells in oeRNaseH1 condition, shows drastic reduction in S phase cells (Fig. 1e). It is therefore possible that cells with oeRNaseH1 activates checkpoint and cells mainly get stuck in early S phase whereas the damage mainly comes from later stages of S phase. To rule out this possibility, authors should: a) show cell cycle profile of oeRNH1 cells by FACS. B) confirm if there is checkpoint activation in oeRNH1 cells. c) also show western blot comparing the level of RNaseH1 expression in control and silno80 cells.
2. Similarly, for Fig.2 where transcription inhibitors have been used to show the rescue of fork progression velocity defects in silno80 cells, authors should perform assays as suggested above.
3. The authors should comment on: why RNH1 over-expression that remove R-loops causing significant defects in fork progression rate in control cells? (fig. 2c, compare lane 1 and 2; fig. 2i, compare lane 1 and 3)
4. Discrepancy between the datasets showing extent of rescue in fork progression defects in silno80 upon oeRNH1. (Compare lane 1 and 4 in fig. 2c and fig. 2i)
5. Further, authors claim that depletion of Ino80 causes asymmetry of forks (89%) (see Pg 11, line 10). However, if the effect on the fork symmetry is so drastic it should show up already in the data from fig. 2. For ex, a violin plot of the IdU tract should show a bimodal distribution of the IdU

tracks in the absence of Ino80. Furthermore, what they claim to be asymmetry could also be broken fibers. Therefore, when it comes to fork asymmetry the whole DNA staining of fibers should be shown to rule out this possibility.

6. Fig. 3b shows rescue in fork asymmetry in silno80 upon oeRNH1 but it does not seem like fork progression defects is rescued as claimed in Fig. 2

7. Contrary to previous reports, the authors claim that chromatin relaxation using HDAC inhibitors play important role in preventing R-loop accumulation and mediating fork progression in silno80 cells, as shown by an epistasis between SAHA/Vorinostaat and oeRNH1. However, it has been well documented from various labs that knock down of HDACs or SAHA/Vorinostaat treatment (with similar conc. that authors used) causes accumulation of R-loops (Salas-Armenteros et al. 2017, EMBO) as well as slows down fork progression rate and activates dormant origins to fire (Conti et al. 2010, Cancer Res.; Bhaskara et al. 2013, Epigenetics and Chromatin) etc. So, in case where Ino80 is absent to resolve R-loops causing replication defect, I would expect HDACi to enhance this phenotype. The authors should comment on this and show S9.6 staining upon treatment with HDACi in their cell lines.

8. In Pg 12, line 11-13; authors claim to analyze R-loops in EdU positive and negative cells and that INO80 depletion increases accumulation of R-loops in cells undergoing DNA replication. With this they conclude that Ino80 suppresses the transcription-replication conflicts, however, from there data fig.4e it is clear that even non-replicating cells show significant increase in R-loops. Moreover, the no. of cells quantified for EdU positive seems to be much lower than EdU negative cells (n for no. of cells for each lane should be mentioned). The authors should discuss the other possibility or as suggested below in comment 9.

9. From tethering experiments (Fig. 7), authors claim that Ino80 promotes resolution of R-loops. However, this whole dataset doesn't discriminate between replicating and non-replicating cells, suggesting that Ino80 can resolve R-loops even in non-replicating cells in contrary to what authors have claimed in the whole manuscript. Therefore, based on data from Fig. 4 and 7, the authors should not stress that the effects are limited to replicating cells, indeed the DNA damage is seen mainly in S phase which could be due to the unresolved R-loops from other phase of cell cycle can cause impediment in fork progression and may cause fork collapse resulting into DNA damage in replicating cells.

10. There are major concerns regarding tethering experiments and the usage of RBD-DsRed protein to signify R-loop. a) The signal between eGFP-lacI and S9.6 staining doesn't colocalize but shows at proximity (fig. 7b) whereas RBD-DsRed and eGFP-LacI colocalize almost completely (Fig. 7e-f). As there is no citation for the usage of RBD-DsRed for recognition of R-loops from previous reports, the authors need to first validate their system by showing colocalization between S9.6 and RBD-DsRed signal in these cells.

11. The authors claim that tethering of Ino80E fusion protein (eGFP-LacI-Ino80E) to lacO does not prevent the R-loop formation but help in its removal. However, looking at the figure pictures 7f (time series) there is no obvious RBD-DsRed focus formation even from time 0min. The time frames also don't make sense when comparing figures 7f and 7g.

Minor Comments:

1. Comment on discussion, Pg 23 line 6-11: Keeping in mind that authors performed this study using cancer cells which shows enhanced transcriptional activity overall, it is likely that Ino80 becomes essential to maintain genome stability due to this high transcriptional activity also at non-coding region that may also favor R-loop formation. Therefore, I would tone down the discussion a bit without generalizing the effects for normal cells with controlled transcriptional activity.

2. In Discussion Pg 22, line 21-22; authors mention "This strongly suggests that INO80 and R-loops co-occupy the same regions of the genome". This is an overstatement as there data from Fig. 5 shows at most or even less than 50% times of overlap between Ino80 and S9.6 foci.

3. To confirm that lacO site enriches for R-loop, authors can perform DRIP with S9.6 antibody to show enrichment of signal above background at lacO site (fig. 7).

4. Fig. 7b shows mostly cytoplasmic S9.6 staining. To validate perhaps show colocalization with

RNaseH-dead mutant.

5. In Fig. 7c, how many cells are quantified per experiment to show S9.6 signal intensity at lacO site?
6. In figure legend of Fig. 1d, please state clearly what is normalized and how.
7. The authors should mention the conversion formula for tract lengths measurements from microns to Kb they used in Fig. 2, in material and methods.
8. There is no mention in text or figure legend of which tracts were measured in (fig. 2b-f), CldU or IdU or both. It only got clear from Pg 10, line 15-16 where it is mentioned "the length of IdU tracks was quantified, as previously".
9. Use consistent nomenclature style, such as a) Ino80 sometimes is in all upper-case letters but not other times, for ex- Fig. 1-2 vs Fig. 3-4. b) RNaseH1 in figures is sometimes oERNH1 and other time oERNaseH1 for ex- Fig. 2 vs Fig. 3 etc.c) similarly in figures, siRNA is mentioned as esiRNA the other times not.

Reviewer #3 (Remarks to the Author):

Pendergast et al NCOMMS-19-31246

This interesting manuscript looks at how the ATP-dependent chromatin remodeller INO80 promotes the resolution of R-loops to prevent transcription-replication conflicts and attendant DNA damage in cancer cells. Briefly, they show that depleting INO80 leads to an increase in R-loops and this is associated with a slight increase in DNA damage; inducing over-expression of RNaseH1 or inhibiting transcription reversed this. Depleting INO80 also leads to an increase in stalled replication which is rescued by overexpressing RNaseH1 suggesting R-loops underlie the stalling. This is supported by the increased signal attributed to R-loops by immunofluorescence. They try and make a direct link between the localisation of INO80 and R-loops using super resolution microscopy and also explore somewhat superficially the chromatin environment of these using data sets for INO80 by CHIP and R-loops by S9.6 DRIP in mouse ES cells. By artificially tethering INO80 to a LacO tandem array that is associated with the presence of R-loops, using an inducible LacI fusion construct they show that induction of INO80 expression is associated with a reduction in R-loop signal. Finally in a variety of cancer cells depletion of INO80 led to a growth defect which was rescued by over expression of RNaseH1. Reasoning that in cancer cells repair of DNA damage due to R-loops is dependent on the base excision repair pathway they inhibited this pathway and showed this led to a marked increase in cell lethality.

Particular points to be addressed

1. The HU data in Figure 1 should be moved to supplementary data as it adds very little to the story but rather detracts by unnecessarily complicating the figure.
2. The knockdown of INO80 should be done using 2 different siRNAs to reduce the likelihood that the effect of treatment is due to an off target action.
3. The data in Figure 2 does not distinguish between slowing of DNA polymerase and an increase in replication fork stalling, The authors acknowledge this by then looking at fork asymmetry in Fig 3 which is much better at assessing fork stalling but they only look at INO80 depletion +/- RNaseH1 overexpression in this way. They should repeat the other experiments in Fig 2 by this method to assess how transcription and histone deacetylase inhibition affect stalling of replication when INO80 is depleted. Later they argue that Fig 2 points to a general reduction in fork speed but this is not persuasive. To do so they need to excluded the contribution of fork stalling. One way to do this would be to score the length of the nascent CldU labelled tracts in fibres which also have the subsequent IdU label. The presence of the latter label would reduce the likelihood that stalling had occurred.

4. The statistical analysis comparing fork rates of control cells +/- Vorinostat (Fig 2i lanes 1 and 5) is referred to in the text but not shown in the figure.

5. In Fig 4b the S9.6 signal is diffuse and quite unlike the punctate signal in 4c - why?

6. The dotblot in Suppl Fig 4b needs repeating ($n \geq 3$) and quantifying.

7. They conclude from Fig 4e that depletion of INO80 led to increased R-loops in cells undergoing DNA replication. The figure to my eye appears to show an increase in R-loops in cells depleted for INO80 whether they are in S-phase or not. However I'm not sure what comparison they are trying to make here. Is it the level of R-loops between EdU+ and EdU -ve cells if so the figure doesn't statistically compare these groups.

8. The colocalisation of INO80 and R-loops by super resolution microscopy would have been strengthened considerably by comparing in more detail the localisation of INO80 by CHIP and R-loops by S9.6 DRIP. Rather than just showing a plot showing some correlation between the signal examples browser tracks would have been illuminating. It is curious that such a low fraction of INO80 is associated with chromatin.

9. The LacO/LacI system is artificial and in order to see if this truly reflects how INO80 affects R-loops they should look at endogenous sites associated with INO80 and R-loops by looking at such loci identified in the relevant data sets for mouse ES cells. After knocking down INO80 in these cells they should look using s9.6 DRIP/QPCR to see if R-loop levels increase.

Reply to Reviewers' comments:

Reviewer #1 (Remarks to the Author):

Reviewer 1(R1). This study reports that INO80 complex is involved in removing R-loops to prevent replication stress in human cells. RNAi depletion of INO80 subunit of this complex leads to increase in signal and to markers of replication stress such as replication fork slowing and gamma-H2AX. Ectopic overexpression of RNase H1 can rescue these phenotypes, as well as some of the lethal effect of depleting INO80 in cancer cell lines. There is some evidence of genome-wide co-localization of R-loops and INO80, and that INO80 suppresses R-loop formation in cis.

On the positive side, this topic is timely and interesting. Firstly, knowledge on INO80 function in replication stress is limited and new insights will greatly contribute to advancing the field. Secondly, this study also uses some potentially interesting new approaches for R-loop detection.

On the negative side, the work as it stands seems quite superficial. The experiments lack controls, leading to not very convincing data. Furthermore, the effect of INO80 depletion on slowing down of replication fork speeds and cell death is much more pronounced than the effect of INO80 depletion on R-loop signal. This suggests that there are other, more important, mechanisms besides R-loops at play. There clearly appears to be some interplay between INO80 and RNaseH1, but mechanistically, the effects could be quite indirect, for example as a result of more DNA damage or stress generally in the absence of INO80.

The data as presented cannot clarify that question. Furthermore, none of the other subunits of the INO80 complex have been depleted to see whether this causes similar phenotypes. The data would be much more convincing with such a broader approach.

Reply:

In our manuscript, we provide evidence that the effect of INO80 depletion in replication and cell death is dependent on the presence of R-loops. We further show that the R-loop signal is increased in INO80-depleted cells, at levels comparable to what has been observed in published studies for R-loop regulators. R-loops have a well-established role in inducing replication stress and DNA damage if not timely resolved. The relationship between DNA damage and cell homeostasis is not linear. The signal from a damaged DNA site is magnified and transduced to other cellular pathways, leading to an overall slowdown of genome replication, induction of cell cycle arrest and, if irreparable, trigger cell death. It is therefore expected that even if a small amount of DNA damage at forks encountering persistent R-loops in the absence of INO80, this can lead to gross aberrations in cell homeostasis.

Reviewer 1 questions whether increased DNA damage could be the reason underlying DNA replication fork stalling in the absence of INO80. In the revised manuscript we addressed the possibility that DNA damage might be responsible for defective sister fork progression. Using an antibody against ssDNA, we tested whether the DNA fibres are intact at sites of asymmetrical sister forks in cells depleted of INO80. Our results, as presented in the revised manuscript show that DNA fibres are intact at asymmetrical forks in siINO80 (Supplementary Fig. 4). This rules out the possibility that the R-loop-dependent defect in replication fork symmetry in siINO80 cells is a result of breaks of DNA fibres.

In the revised manuscript we provide additional controls and new experiments that further substantiate our proposed model that the role of the INO80 complex in promoting DNA replication is associated with R-loops. The new experiments are presented below in our Replies to the respective Specific Points.

R1(cont). Finally, the study is using cancer cell lines, but that is not the same as directly investigating whether INO80 is involved in response to oncogenes. Therefore, the impact of the findings is not as high as it might be suggested in the introduction.

Reply:

We agree with the reviewer and apologize for the confusion. In the revised manuscript we have removed the oncogene part from the introduction. The potential role of INO80 in oncogenic transcription and DNA replication in cancer cells by R-loop regulation is now discussed in the Discussion section (page 29, lines 6-8).

R1. Specific points:

- There are no controls for off-target effects on the INO80 siRNA. It is reported that the esiRNA approach used here is more specific and with fewer off-target effects, but how sure can we be of this? It would be much better to use two different siRNA sequences to target INO80 or to rescue effects by INO80 re-expression.

Reply:

To address the reviewer's concern, we have conducted in the revised manuscript the following new experiments and controls:

- (i) Rescue of the replication defect of siINO80 cells by ectopic expression of siRNA immune INO80 cDNA (Supplementary Fig. 2a-d). This indicates that the observed replication defect is specifically due to INO80 loss.*
- (ii) Demonstrate that depletion of the INO80 subunit ACTR8/ARP8 leads to DNA replication defects and that the siACTR8-dependent DNA replication defect is rescued by RNase H1 overexpression (Supplementary Fig. 3e-h), similarly to what we observe when INO80 is depleted.*
- (iii) Demonstrate that depletion of INO80 by 4 independent lentiviral shRNAs leads to increased accumulation of R-loops compared to control cells (Fig. 4f-g).*

These results strongly indicate that the observed effects in R-loops and DNA replication are due to loss of INO80 function.

R1. - Figure 4: The effects of INO80 depletion on R-loop signals are quite moderate and these experiments are lacking controls. Specifically, it would be good practice to use treatment with recombinant RNaseH as controls for the S9.6 quantification in Fig. 1b.

Reply:

Our results show increased R-loop signal in INO80-depleted cells at levels comparable to what has been reported for depletion of known R-loop regulators (for examples see ¹⁻³). Our analysis indicates that the increase in accumulation of R-loops in the absence of INO80 is statistically significant (Fig. 4d, e). Importantly, the increase in R-loops in the absence of INO80 is associated with an increase in genomic instability and defective proliferation. These results strongly suggest that the role of INO80 in counteracting accumulation of R-loops is not moderate but biologically important.

Following the reviewer's suggestion, we now show that recombinant RNaseH 1 treatment removes the S9.6 signal (from R-loop enrichment) in cells that are depleted for INO80 (Supplementary Fig. 7d). We further show that the increase in DRIP-qPCR signal observed at the beta-actin and EGR1 genes upon INO80 depletion is diminished upon treatment of the genomic DNA with recombinant RNaseH1 prior to DRIP (Supplementary Fig. 7f).

R1. - Fig. S4b: The S9.6 slot blot experiment needs controls such as DNA loading control and RNaseH

treatment as well. Ideally, other methods of R-loop quantification should also be used, such as DRIP qPCR with the appropriate controls, to further strengthen the argument.

Reply:

Per the reviewer's request, we now show by DRIP-qPCR that the enrichment of R-loops at the beta-actin and EGR1 genes is increased upon INO80 depletion (Fig. 4g). The increase in R-loops observed in the INO80 depleted cells is diminished upon treatment of the genomic material with recombinant RNaseH1 prior to the S9.6 IP (Supplementary Fig. 7f).

As DRIP analysis is now included in the paper, the slot blot experiment does not provide any added value or additional information to the story and it has therefore been removed from the manuscript.

R1. - Figure 5: The high-resolution microscopy data very interesting, but results are only shown for 4 cells. Is this staining and co-localisation really that clear in the majority of cells and are the data representative? These results would also need more verification by other methods. If INO80 co-localises with R-loops, this should also show up in standard DRIP qPCR experiments. For example, it would be worth using ChIP and DRIP to test for INO80 enrichment at known R-loop loci such as across the beta-Actin gene (K. Skourti-Stathaki Mol Cell 2011) and at some of the loci that have been identified to contain R-loops and INO80 in the analysis in Figure 6C.

Reply:

In STED super-resolution microscopy, we are measuring and plotting data from a huge number of particles. The data pooled together in 5d,e,f are extremely significant ($P < 0.0001$). The images shown and data analysed are representative. Following the reviewer's request, we have analyzed more cells (Fig. 5c and Supplementary Fig. 8). The results from the new co-localization analysis are consistent with our previous analyses. Importantly, we would like to emphasize that the STED data (R-loop intensity, volume, length and % of R-loops colocalization with INO80) are highly statistically significant ($P < 0.0001$) across all cells.

As suggested by the reviewer we now show by analysis of ChIP-seq and DRIP-seq genomic data that INO80 and R-loops are co-enriched at several specific loci, including the beta-actin gene (Fig. 4g). This new analysis further supports our conclusion for a genome wide association between INO80 and R-loops.

R1. - Figure 7: The imaging data using LacI tethering are very interesting, but is it possible that the larger fusions with RNH1 or INO80 sterically prevent binding of the RBD-dsRed? It looks as if RBD-dsRed may be not binding properly in presence of the longer construct. This assay would need a control experiment with a protein fusion that cannot remove R-loops, such as catalytically inactive RNaseH1 mutant.

Reply: We agree that the control suggested by the reviewer is a good control for the RBD-dsRed experiment. We cloned the catalytically dead RNaseH1 in the lacI plasmid, but the lab was shut down due to COVID-19 before we could finish the experiment.

While we cannot formally exclude that RBD binding is defective in the presence of LacI-INO80E, our results obtained from the kinetics analysis conducted at the lacI site (Fig. 7d-g) argue against this possibility. If RBD-dsRed binding was adversely affected by lacI-INO80E, and the resolution kinetics remained the same upon binding of either lacI-GFP or lacI-INO80E, we would expect to see an increase in negative values in the LacI-INO80E compared to lacI-GFP. However, this is not the case, as we observed similar negative values in the lacI-GFP and lacI-INO80E cells (Fig. 7g). Therefore, our results contest the hypothesis that defective RBD-dsRed binding is responsible for the changes in the intensity of RBD-DsRed at the lacO site observed upon binding of the lacI-INO80E. The possibility of

defective RBD binding in the presence of INO80 is argued in the Discussion section (page 27, lines 9-15).

R1. - Figure 7f: What are the two different “merge” panels?

Reply:

The original merge panels were to show (i) the LacI-GFP and RBD-Ds-Red foci in the whole cell and ii) a zoom-in on the LacI-GFP/RBD-Ds-Red foci in the same cells. For clarity, we have now moved the whole cell images from the main figure and placed them into supplemental material (Supplementary Fig. 11d).

R1. - The discussion states that “chemical inhibition of transcription rescued almost entirely the defects in DNA replication progression and in fork symmetry observed in INO80-depleted cells during normal S phase (Fig. 2 and 3).” But the effect of chemical transcription inhibitors in Fig. 2f is actually quite small.

Reply:

We apologize for the mistake. We have corrected the specific part in the Discussion and we now state that “...the replication-associated DNA damage of INO80-depleted cells was significantly relieved by overexpression of the RNA:DNA endonuclease RNase H1 or chemical inhibition of transcription...” (Page 24, line 18-20).

Reviewer #2 (Remarks to the Author):

R2. In this manuscript by Prendergast and colleagues, “Resolution of R-loops by INO80 promotes DNA replication and maintains cancer cell proliferation and viability”, the authors set out to address how cancer cells manage to resolve R-loops that form obstacle to replication fork progression and further show that the suppression of r-loops by INO80 maintains proliferation of cancer cells. To this end, the authors mainly used RNaseH over-expression system for rescue experiments to show that R-loops accumulation is the main source of replication-associated DNA damage in Ino80 depleted cancer cells. Further, they performed DNA fibre analysis to observe replication fork progression defects in silno80 cells. From this they conclude that accumulation of R-loops is the main cause of replication defects, which also leads to fork asymmetry and both of which can be rescued by either over-expression of RNaseH1 enzyme or by using HDAC inhibitors or by using global transcription inhibitors.

Additionally, they generated lacO-lacI system to study the R-loop resolution at lacO site. They generate fusion proteins of Ino80E subunit with LacI for artificial tethering of Ino80 complex to the lacO site and generate truncated RNA binding domain of RNaseH1 (RBD-DsRed) to signify R-loops in live-imaging assays. Using this system, they claim to show that artificial tethering of Ino80 helps in efficiently resolving R-loops rather than preventing them from forming at first place. Furthermore, the authors show that depletion of Ino80 results in synthetic lethality in multiple cancer cell lines, which is due to accumulation of unresolved R-loops and can be rescued by RNaseH1 over-expression.

Although the premise of work is interesting and some of the findings being novel however some key conclusions have been based on data which is not very convincing. For example- it is conceptually hard to accept the fact that the transcription inhibitors and HDAC inhibitors that enhances transcription in general, are rescuing the same replication defects shown for silno80 cells. Whereas HDAC inhibitors have been suggested to accumulate R-loops and slow down replication fork progression, previously (see more details below).

Furthermore, the claim that Ino80 resolves R-loops in replicating cells mainly, is not strongly supported by their data.

Other than this, there are frequent discrepancies in their own data within or in between figures, which has not been addressed by the authors. Overall, the work is interesting but requires addressing key issues with appropriate experiments with validations, to be considered for publication.

Reply:

We apologize if the description of our results led inadvertently to the mistaken impression that INO80 resolves R-loops in replicating mainly cells. In our manuscript we do not claim that Ino80 resolves R-loops in replicating cells mainly. Rather, as our data indicate, depletion of INO80 led to enhanced accumulation of R-loops in cells both outside and inside S phase. To avoid any further misunderstanding, we have changed the text accordingly (page 13, lines 8-10).

R2. Major comments:

R2. 1. In Figure 1, authors claim that HU-mediated replication stress causes DNA damage, shown by γ -H2AX intensity and % of cells, in replicating (S phase) cells and not in outside of S phase cells upon knockdown of Ino80. This they claim is due to accumulation of R-loops as this was rescued by over-expression of RNaseH1 which resulted in significantly reduced, both the intensity and the percentage of γ H2AX positive cells (Fig. 1c-e). However, the percentage of S phase cells in both control and silno80 cells in oeRNaseH1 condition, shows drastic reduction in S phase cells (Fig. 1e).

Reply:

In our paper we did not claim or show that the percentage of S phase cells in both control and silno80 cells in oeRNaseH1 condition, shows drastic reduction in S phase cells (old Fig. 1e, new Supplementary Fig. 5e). The data in old Fig 1e showed a reduction in the number of γ H2AX-positive cells in oeRNaseH1 condition. These data are presented now as Supplementary Fig. 5e. We apologize for any confusion our data presentation led to.

R2cont. It is therefore possible that cells with oeRNaseH1 activates checkpoint and cells mainly get stuck in early S phase whereas the damage mainly comes from later stages of S phase. To rule out this possibility, authors should: a) show cell cycle profile of oeRNH1 cells by FACS. B) confirm if there is checkpoint activation in oeRNH1 cells. c) also show western blot comparing the level of RNaseH1 expression in control and silno80 cells.

Reply:

Per the reviewer's request we conducted FACS analysis and evaluated checkpoint activation by analyzing the levels of phosphorylation of Chk1-S345 – a downstream target of ATR, by western. In our revised manuscript we show that cells depleted for INO80 are enriched in early S-phase and have activated the S-phase checkpoint, as indicated by increased pChk1 levels (Supplementary Fig. 3b, c and Fig. 3b respectively). These results rule out the possibility that the damage in the absence of INO80 comes mainly from later stages of S-phase. Over-expression of RNase H1 slightly increases the number of cells in early S-phase compared to control cells and partially activates the S-phase checkpoint levels (Supplementary Fig. 3b, c and Fig. 3b respectively). This is not unexpected since it has been reported that over-expression of RNaseH1 can lead to destabilization of RNA-DNA hybrids around a double strand break, delaying DSB repair⁴ and to cleavage of the RNA primer from Okazaki fragments⁵. However, RNase H1 overexpression rescues the cell cycle defect of silINO80 cells and concomitantly alleviates the S-phase cell cycle checkpoint levels (Supplementary Fig. 3b, c and Fig. 3b

respectively). These results indicate that INO80 and RNase H1 act in early S-phase and suggest that INO80 counteracts R-loops to suppress DNA damage inside S-phase.

With regard to the western for RNaseH1, we were unable to complete the experiment because our labs were shut down due to COVID-19. Nevertheless, our data demonstrate that RNaseH1 ectopic expression reverses the phenotypes of INO80kd to control levels and that control and siINO80 cells overexpressing RNaseH1 have similar FACS profiles and CHK1 phosphorylation levels (Fig. 3a, Supplementary Fig. 3b-d). Therefore, any potential difference in expression of RNaseH1 in the two cell lines is not going to affect the interpretation of our data.

R2. 2. Similarly, for Fig.2 where transcription inhibitors have been used to show the rescue of fork progression velocity defects in silno80 cells, authors should perform assays as suggested above.

Reply:

Our results in old Fig. 2 (new Fig. 1 in the revised manuscript) indicate that co-transcriptional R-loops are an obstacle for fork progression in the absence of INO80. Global inhibition of transcription by chemicals leads rescues the replication defect in siINO80 cells and to a reduction of R-loops across the genome as a secondary result of decreased transcription. Our DNA fibre experiments in the presence of chemical transcription inhibitors demonstrate that contrary to RNaseH1 over-expression, the DNA replication rates are not affected by the inhibitors in the concentration used. This indicates that chemical transcription inhibition does not impede DNA synthesis. Therefore, the experiments suggested by the reviewer would add no new information in regard to the role of Ino80 in promoting DNA replication.

R2. 3. The authors should comment on: why RNH1 over-expression that remove R-loops causing significant defects in fork progression rate in control cells? (fig. 2c, compare lane 1 and 2; fig. 2i, compare lane 1 and 3)

Reply:

As mentioned above, it has been reported that overexpression of RNaseH1 can lead to destabilization of RNA-DNA hybrids around a double strand break, delaying DSB repair⁴ and to cleavage of the RNA primer from Okazaki fragments⁵. We have observed a partial activation of the intra-S phase checkpoint upon RNase H1 overexpression (Fig. 3a), in agreement with the reported phenotypes. It is therefore likely that slower replication rates upon RNaseH1 overexpression is associated with the partial activation of the S-phase checkpoint. This is now mentioned in the text (page 11, lines 7-9).

R2. 4. Discrepancy between the datasets showing extent of rescue in fork progression defects in silno80 upon oeRNH1. (Compare lane 1 and 4 in fig. 2c and fig. 2i)

Reply:

We thank Reviewer 2 for pointing out this discrepancy. This was due to a deviation observed in one of the experiments in the old Figure 2i set, which skewed the final result. To address this point, we conducted one more experiment, the results of which are now presented in the new Figure 1i. The results of the new experiment are in agreement with our previous results and confirm that the extent of rescue in fork progression defect in siINO80 upon oeRNH1 in Figure 1i is similar to Figure 1f.

R2. 5. Further, authors claim that depletion of Ino80 causes asymmetry of forks (89%) (see Pg 11, line 10). However, if the effect on the fork symmetry is so drastic it should show up already in the data from fig. 2. For ex, a violin plot of the IdU tract should show a bimodal distribution of the IdU tracks in the absence of Ino80. Furthermore, what they claim to be asymmetry could also be broken fibers.

Therefore, when it comes to fork asymmetry the whole DNA staining of fibers should be shown to rule out this possibility.

Reply:

As requested by the reviewer, ssDNA staining for old Figure 3 (new Figure 2) was conducted. Our results show that asymmetrical forks in siINO80 are found at intact DNA fibres (Supplementary Fig. 4). This result rules out the possibility that the R-loop-dependent defect in replication fork symmetry in siINO80 cells is a result of DNA damage.

Here, we would like to note that, purely experimentally, triple staining of DNA fibres made the analysis of the results challenging. We observed that ssDNA staining was often inhibited in the proximity of CldU/IdU patches, likely due to a local drops in secondary antibody concentration by competitive binding to anti-IdU antibody (Both anti-IdU and anti-ssDNA antibodies are mouse and staining is sequential – first anti-IdU and AF488 secondary followed after extensive washes by anti-ssDNA Ab and AF647 secondary.) In several instances, we observed that tracks which were clearly on the same fibre as judged by proximity and stretching failed to display proper ssDNA staining. Nevertheless, our analysis of intact DNA fibres in control and siINO80 cells clearly showed that asymmetrical forks in the absence of INO80 is not a result of DNA damage (Supplementary Fig. 4).

In regard to the violin plot question: Co-transcriptional R-loops are dynamic structures mainly enriched in proximity to transcription start sites (promoters) or at transcription termination sites of poly(A)-dependent genes⁶. Replication also preferentially initiates at the transcription start site of genes occupied by high levels of RNA polymerase II, which favors formation of R-loops⁷. R-loop are rare structures occupying up to 5% of mammalian genomes⁶, making encounters between forks and R-loops rare events during DNA replication. Thus, as forks replicating over R-loop regions account only for a small fraction of the DNA replication events captured experimentally by the DNA fibre assay (new Figure 1), the rarity of these events is not expected to lead to a significant change in the distribution of IdU tracts in a violin plot analysis. This is evident in the violin plot analysis in control and siINO80 cells that we conducted and presented below. In agreement with our results in new Figure 1, our violin plot analysis indicates both a significant skewing and some bimodal distribution of idU tracts in the siINO80 cells.

Figure legend: Violin plot analysis of the distribution of the second (green) label track lengths control (siGFP) and INO80-deficient cells (siINO80) from Figure 1e. Boxplots of the same measurements are overlaid. At least 150 measurements were made per condition. $P \leq 0.0001$.

R2. 6. Fig. 3b shows rescue in fork asymmetry in silno80 upon oeRNH1 but it does not seem like fork progression defects is rescued as claimed in Fig. 2

Reply:

In the revised version, (old) Fig. 3b, c (new Fig. 2b, c), it can be clearly seen that the majority of the arms 1 and 2 (Figure 2c, x and y axes) in silINO80 upon oeRNH1 cells is longer than the arms in silINO80. Therefore, oeRNH1 rescues the fork progression defect of INO80kd, similar to Figure 1.

R2. 7. Contrary to previous reports, the authors claim that chromatin relaxation using HDAC inhibitors play important role in preventing R-loop accumulation and mediating fork progression in silno80 cells, as shown by an epistasis between SAHA/Vorinostaat and oeRNH1. However, it has been well documented from various labs that knock down of HDACs or SAHA/Vorinostaat treatment (with similar conc. that authors used) causes accumulation of R-loops (Salas-Armenteros et al. 2017, EMBO) as well as slows down fork progression rate and activates dormant origins to fire (Conti et al. 2010, Cancer Res.; Bhaskara et al. 2013, Epigenetics and Chromatin) etc. So, in case where Ino80 is absent to resolve R-loops causing replication defect, I would expect HDACi to enhance this phenotype. The authors should comment on this and show S9.6 staining upon treatment with HDACi in their cell lines.

Reply:

We apologize for not clearly articulating our results. Contrary to the Reviewer's comment, we do not claim "...HDAC inhibitors play important role in preventing R-loop accumulation and mediating fork progression in silno80 cells" in our manuscript. In new Figure 1i (old Fig. 2i), we show that SAHA treatment reduces replication rates relative to untreated cells (Figure 1i, lanes 1 and 5), as expected. Therefore, our results on SAHA/vorinostat are consistent with the current literature.

Importantly, our data indicate that the defect in replication upon HDAC inhibition is not related to the defect in replication in the absence of INO80. This is not surprising given that histone acetylation has both positive and negative effects in genome function and stability, depending on which of the histone lysine residues are being modified and SAHA is a global HDAC inhibitor, acting on a broad range of class I, II and IV of histone deacetylases and subsequently leading to indiscriminatory histone acetylation.

Therefore, our results suggest that although inhibition of specific HDACs has a detrimental effect on DNA replication that is independent of INO80, inhibition of other HDACs by SAHA relieves the replication defect of INO80 depletion. This is not contradictory to previous reports but expands our knowledge on HDACs.

Based on our data, the conclusion from our vorinostat/SAHA experiment (Figure 1i) as stated in the manuscript is: "Taken together our results suggest that chromatin regulation by INO80 plays an important role in preventing co-transcriptional R-loops from interfering with replication fork progression." (Page 9, lines 14-16).

We have now addressed this point by providing a more detailed description of the results of Figure 1i in the revised manuscript (Pages 8, line 23 to page 9, line 14).

R2. 8. In Pg 12, line 11-13; authors claim to analyze R-loops in EdU positive and negative cells and

that INO80 depletion increases accumulation of R-loops in cells undergoing DNA replication. With this they conclude that Ino80 suppresses the transcription-replication conflicts, however, from there data fig.4e it is clear that even non-replicating cells show significant increase in R-loops. Moreover, the no. of cells quantified for EdU positive seems to be much lower than EdU negative cells (n for no. of cells for each lane should be mentioned). The authors should discuss the other possibility or as suggested below in comment 9.

Reply:

We apologize if the description of our results led inadvertently to the mistaken impression that INO80 resolves R-loops in replicating mainly cells. In our manuscript we do not claim that Ino80 resolves R-loops in replicating cells mainly. Rather, as our data indicate, depletion of INO80 led to enhanced accumulation of R-loops in cells both outside and inside S phase. To avoid any further misunderstanding, we have changed the text accordingly (page 13, lines 8-10).

*The number n of EdU positive and negative cells are now mentioned in the legend of Fig. 4e. In both EdU negative and positive cells, the increase in R-loop intensity in siINO80 compared to siNT cells is significant (**).*

R2. 9. From tethering experiments (Fig. 7), authors claim that Ino80 promotes resolution of R-loops. However, this whole dataset doesn't discriminate between replicating and non-replicating cells, suggesting that Ino80 can resolve R-loops even in non-replicating cells in contrary to what authors have claimed in the whole manuscript. Therefore, based on data from Fig. 4 and 7, the authors should not stress that the effects are limited to replicating cells, indeed the DNA damage is seen mainly in S phase which could be due to the unresolved R-loops from other phase of cell cycle can cause impediment in fork progression and may cause fork collapse resulting into DNA damage in replicating cells.

Reply:

We apologize for the confusion. As stated above, we never claimed in the original manuscript that "the effects (of INO80 depletion in R-loop accumulation) are limited to replicating cells". Our result demonstrating that loss of INO80 leads to increase in R-loops in replicating and non-replicating cells (Figure 4e) agrees with the reviewer's point that Ino80 can resolve R-loops in both replicating and non-replicating cells. To avoid any further misunderstanding, we have changed the text accordingly (page 13, lines 8-10).

R2. 10. There are major concerns regarding tethering experiments and the usage of RBD-DsRed protein to signify R-loop. a) The signal between eGFP-lacI and S9.6 staining doesn't colocalize but shows at proximity (fig. 7b) whereas RBD-DsRed and eGFP-LacI colocalize almost completely (Fig. 7e-f). As there is no citation for the usage of RBD-DsRed for recognition of R-loops from previous reports, the authors need to first validate their system by showing colocalization between S9.6 and RBD-DsRed signal in these cells.

Reply:

In Figure 7b, the eGFP-lacI (green) is directly overlapping the S9.6 staining (red). This colocalisation is clear, as a yellow signal, which indicates overlap between red and green signals, can be observed inside the green focus in the "Merge" panel (Fig. 7b). There is a second red 'double-dot' in the red S9.6 that is clearly outside the eGFP-lacI, and therefore is not relevant.

Importantly, in our signal quantification analysis for Fig. 7, we exclusively quantified the S9.6 and the RBD-DsRed signals overlapping with the eGFP-LacI signal as stated in the text (page 18, line 15 and page 19, line 11).

In regard to the usage of the RNA Binding Domain (RBD) of RNase H1 for recognition of R-loops, the RBD construct has been well documented and verified as an alternative method to S9.6 for in-vivo recognition of R-loops². The RBD-DsRed is a variant of the RBD-GFP construct referred as HB-GFP in ², which is cited in our manuscript.

R2. 11. The authors claim that tethering of Ino80E fusion protein (eGFP-LacI-Ino80E) to lacO does not prevent the R-loop formation but help in its removal. However, looking at the figure pictures 7f (time series) there is no obvious RBD-DsRed focus formation even from time 0min. The time frames also don't make sense when comparing figures 7f and 7g.

Reply:

Fig. 7f is a 48-minute time series from the 1500 minutes experiment. There is a formation of RBD-DsRed focus even at time 0 as indicated by the yellow signal from the overlap of green and red channels in the merge panel.

The RBD-DsRed focus is evident under the GFP-LacI-Ino80E in Fig. 7f, as yellow signal, which indicates overlap between red and green. RBD-DsRed can be observed inside the green focus in the time-frames t(12), t(30) and t(48) (Fig. 7f, Merge).

We apologize for the confusion caused by the original Fig. 7g. In the revised manuscript we present a new analysis in Fig. 7g that shows the Fold Change in the intensity of the DsRed signal under the eGFP signal between two time points during the course of the experiment (n=approximately 900 values). The new analysis provides evidence against the possibility that tethering of INO80 at the LacO locus prevents R-loop formation. This is now discussed in detail in the text (page 20, lines 9-16).

R2. Minor Comments:

R2. 1. Comment on discussion, Pg 23 line 6-11: Keeping in mind that authors performed this study using cancer cells which shows enhanced transcriptional activity overall, it is likely that Ino80 becomes essential to maintain genome stability due to this high transcriptional activity also at non-coding region that may also favor R-loop formation. Therefore, I would tone down the discussion a bit without generalizing the effects for normal cells with controlled transcriptional activity.

Reply:

We thank Reviewer 2 for the suggestion. The discussion of this part has been modified (page 26, line 21 to page 27, line 3).

R2. 2. In Discussion Pg 22, line 21-22; authors mention "This strongly suggests that INO80 and R-loops co-occupy the same regions of the genome". This is an overstatement as there data from Fig. 5 shows at most or even less than 50% times of overlap between Ino80 and S9.6 foci.

Reply:

This part of the discussion has now been rephrased to avoid any overstatement. We now mention that "STED nanoscopy revealed that up to 50% of the total R-loop sites detected at ~50nm resolution at a given time are bound by INO80 (Fig. 5)" (page 26, lines 2-3)... "although super-resolution imaging on fixed cells by STED does not allow us to ascertain the dynamics or true frequency of the INO80:R-loop interactions, 3D analysis of the STED data revealed a preferential colocalization of INO80 with the largest and most enriched R-loop sites (Fig. 5)." (Page 26, lines 8-11).

R2. 3. To confirm that lacO site enriches for R-loop, authors can perform DRIP with S9.6 antibody to show enrichment of signal above background at lacO site (fig. 7).

Reply: *While this is a good suggestion, it is not possible to conduct quantitative DRIP at the lacO site because of its repetitive sequence.*

R2. 4. Fig. 7b shows mostly cytoplasmic S9.6 staining. To validate perhaps show colocalization with RNaseH-dead mutant.

Reply:

RNaseH-dead mutant was constructed but the lab was shut down due to COVID-19 before we could do the experiment. However, during the revision process we have used recombinant RNase H1 to digest the S9.6 signal in PC3 cells. The loss of both cytoplasmic and nuclear S9.6 signal upon treatment suggests that both signals are R-loops.

R2. 5. In Fig. 7c, how many cells are quantified per experiment to show S9.6 signal intensity at lacO site?

Reply:

The number of cells for the analysis in Fig. 7c is now mentioned in the figure legend.

R2. 6. In figure legend of Fig. 1d, please state clearly what is normalized and how.

Reply:

The legend for new Fig. 3 (old Fig.1) in the revised manuscript states what is normalized and how.

R2. 7. The authors should mention the conversion formula for tract lengths measurements from microns to Kb they used in Fig. 2, in material and methods.

Reply:

The conversion formula is described in the material and methods.

R2. 8. There is no mention in text or figure legend of which tracts were measured in (fig. 2b-f), CldU or IdU or both. It only got clear from Pg 10, line 15-16 where it is mentioned “the length of IdU tracks was quantified, as previously”.

Reply:

This is corrected in the revised manuscript.

R2. 9. Use consistent nomenclature style, such as a) Ino80 sometimes is in all upper-case letters but not other times, for ex- Fig. 1-2 vs Fig. 3-4. b) RNaseH1 in figures is sometimes oeRNH1 and other time oeRNaseH1 for ex- Fig. 2 vs Fig. 3 etc.c) similarly in figures, siRNA is mentioned as esiRNA the other times not.

Reply:

This is corrected in the revised manuscript.

Reviewer #3 (Remarks to the Author):

Pendergast et al NCOMMS-19-31246

This interesting manuscript looks at how the ATP-dependent chromatin remodeller INO80 promotes the resolution of R-loops to prevent transcription-replication conflicts and attendant DNA damage in cancer cells. Briefly, they show that depleting INO80 leads to an increase in R-loops and this is associated with a slight increase in DNA damage; inducing over-expression of RNaseH1 or inhibiting transcription reversed this. Depleting INO80 also leads to an increase in stalled replication which is rescued by overexpressing RNaseH1 suggesting R-loops underlie the stalling. This is supported by the increased signal attributed to R-loops by immunofluorescence. They try and make a direct link between the localisation of INO80 and R-loops using super resolution microscopy and also explore somewhat superficially the chromatin environment of these using data sets for INO80 by ChIP and R-loops by S9.6 DRIP in mouse ES cells. By artificially tethering INO80 to a LacO tandem array that is associated with the presence of R-loops, using an inducible LacI fusion construct they show that induction of INO80 expression is associated with a reduction in R-loop signal. Finally in a variety of cancer cells depletion of INO80 led to a growth defect which was rescued by over expression of RNaseH1. Reasoning that in cancer cells repair of DNA damage due to R-loops is dependent on the base excision repair pathway they inhibited this pathway and showed this led to a marked increase in cell lethality.

Particular points to be addressed

R3. 1. The HU data in Figure 1 should be moved to supplementary data as it adds very little to the story but rather detracts by unnecessarily complicating the figure.

Reply:

The HU data have been moved to supplementary data per Reviewer 3 request (new Supplementary Fig. 5 and Supplementary Fig. 6). Since the work focuses on the role of INO80 in DNA replication, we now show the replication data first to make the presentation clearer.

R3. 2. The knockdown of INO80 should be done using 2 different siRNAs to reduce the likelihood that the effect of treatment is due to an off target action.

Reply:

To address a possible off target action, we have conducted and include in the revised manuscript the following controls and experiments:

- (i) Rescue of the replication defect of siINO80 cells by ectopic expression of siRNA immune INO80 cDNA (Supplementary Fig. 2a-d). This indicates that the observed replication defect is specifically due to INO80 loss.*
- (ii) Demonstrate that depletion of the INO80 subunit ACTR8/ARP8 leads to DNA replication defects and that the siACTR8-dependent DNA replication defect is rescued by RNase H1 overexpression (Supplementary Fig. 3e-h), similarly to what we observe when INO80 is depleted.*
- (iii) Demonstrate that depletion of INO80 by 4 independent lentiviral shRNAs leads to increased accumulation of R-loops compared to control cells (Fig. 4f-g).*

R3. 3. The data in Figure 2 does not distinguish between slowing of DNA polymerase and an increase in replication fork stalling, The authors acknowledge this by then looking at fork asymmetry in Fig 3 which is much better at assessing fork stalling but they only look at INO80 depletion +/- RNaseH1 overexpression in this way. They should repeat the other experiments in Fig 2 by this method to assess how transcription and histone deacetylase inhibition affect stalling of replication when INO80 is depleted.

Reply:

It is well established that transcription inhibition reduces co-transcriptional R-loop formation. It is therefore a logical assumption that chemical inhibition of RNAPII transcription would rescue the R-loop dependent asymmetry in cells depleted of INO80, similar to RNaseH1 overexpression.

We were not able to directly test the effect of HDAC inhibition in fork asymmetry in siINO80 cells as the lab was locked down due to COVID-19. However, we expect that HDAC inhibition would rescue fork asymmetry of siINO80 cells similar to RNaseH1 overexpression for the following reason: Our analysis has indicated an epistatic relationship between RNaseH1 overexpression and HDAC inhibition by Vorinostat in rescuing the DNA synthesis rates in the absence of INO80 (Figure 1i, lanes 4 and 8). Therefore, although we cannot exclude the possibility that HDACi might increase DNA replication rates in siINO80 cells in an R-loop independent way, we believe this is unlikely given that RNaseH1oe and vorinostat rescue the replication defect of siINO80 in an epistatic way. Since persistent R-loops cause replication fork asymmetry in the absence of INO80 (new Figure 2) our results support the possibility that chemical inhibition of histone deacetylation would rescue fork asymmetry caused by INO80 depletion.

R3. 3.cont: Later they argue that Fig 2 points to a general reduction in fork speed but this is not persuasive. To do so they need to excluded the contribution of fork stalling. One way to do this would be to score the length of the nascent CldU labelled tracts in fibres which also have the subsequent IdU label. The presence of the latter label would reduce the likelihood that stalling had occurred.

Reply:

We scored the length of nascent CldU labelled tracts in fibres, which have subsequent IdU label per the reviewer's request. Our analysis indicates that similar to IdU tracts, CldU tracts are also decreased in the absence of INO80 (Supplementary Fig. 3d). This reduces the likelihood that fork stalling contributes to the reduction of DNA synthesis rates observed in old Figure 2 (new Figure 1).

R3. 4. The statistical analysis comparing fork rates of control cells +/- Vorinostat (Fig 2i lanes 1 and 5) is referred to in the text but not shown in the figure.

Reply:

We apologize for the oversight. The statistical analysis is now shown in the new Figure (Fig.1i).

R3. 5, In Fig 4b the S9.6 signal is diffuse and quite unlike the punctate signal in 4c - why?

Reply:

Prior to quantification the images in Fig. 4c were deconvolved (see materials and methods). The image in old Fig. 4b was acquired with a different system (Zeiss AxioImager) and was not deconvolved. To avoid confusion, the non-deconvolved image is now presented in Supplementary Fig 7.

R3. 6. The dotblot in Suppl Fig 4b needs repeating (n>=3) and quantifying.

Reply:

In the revised manuscript we now present DRIP-qPCR data which show that depletion of INO80 leads to increased enrichment of R-loops at specific genomic regions (Fig. 4g). Given that, the dotblot experiment has been removed from the manuscript as it does not provide any added value or additional information to the story.

R3. 7. They conclude from Fig 4e that depletion of INO80 led to increased R-loops in cells undergoing DNA replication. The figure to my eye appears to show an increase in R-loops in cells depleted for

INO80 whether they are in S-phase or not. However I'm not sure what comparison they are trying to make here. Is it the level of R-loops between EdU+ and EdU -ve cells if so the figure doesn't statistically compare these groups.

Reply:

We apologize if the description of our results led inadvertently to the mistaken impression that INO80 resolves R-loops in replicating mainly cells. In our manuscript we do not claim that Ino80 resolves R-loops in replicating cells mainly. Rather, as our data indicate, depletion of INO80 led to enhanced accumulation of R-loops in cells both outside and inside S phase. To avoid any further misunderstanding, we have changed the text to reflect our finding that loss of INO80 leads to increase in R-loops in both replicating and non-replicating cells (page 13, lines 8-10).

R3. 8. The colocalisation of INO80 and R-loops by super resolution microscopy would have been strengthened considerably by comparing in more detail the localisation of INO80 by ChIP and R-loops by S9.6 DRIP. Rather than just showing a plot showing some correlation between the signal examples browser tracks would have been illuminating. It is curious that such a low fraction of INO80 is associated with chromatin.

Reply:

In the revised manuscript, we now present browser tracks from several examples of co-enrichment between INO80 and R-loops at specific genomic regions (Fig. 6b and Supplementary Fig. 10d).

In regard to the cellular fractionation assay, the high salt wash (600mM) extracts soluble histones and nuclear proteins that are either soluble or loosely associated with chromatin, e.g. histone chaperones⁸. Under these conditions, only factors which associate tightly with chromatin or are part of the nucleosome remain in the 'Chromatin' associated fraction. INO80 is a highly abundant protein that belongs to the SWI/SNF family of ATPases, which dynamically associate with chromatin at specific sites across the genome. Therefore, it is expected that most INO80 at any given time is either not associated or, most likely, loosely bound to DNA and as such, it would be released in the high salt wash fraction.

R3. 9. The LacO/LacI system is artificial and in order to see if this truly reflects how INO80 affects R-loops they should look at endogenous sites associated with INO80 and R-loops by looking at such loci identified in the relevant data sets for mouse ES cells. After knocking down INO80 in these cells they should look using s9.6 DRIP/QPCR to see if R-loop levels increase.

Reply:

We thank the reviewers for the suggestion. In our new manuscript we present DRIP-qPCR data which show that depletion of INO80 leads to increased enrichment of R-loops at specific genomic regions (Fig. 4g).

References

- 1 Sollier, J. *et al.* Transcription-coupled nucleotide excision repair factors promote R-loop-induced genome instability. *Mol Cell* **56**, 777-785, doi:10.1016/j.molcel.2014.10.020 (2014).
- 2 Bhatia, V. *et al.* BRCA2 prevents R-loop accumulation and associates with TREX-2 mRNA export factor PCID2. *Nature* **511**, 362-365, doi:10.1038/nature13374 (2014).
- 3 Nguyen, H. D. *et al.* Functions of Replication Protein A as a Sensor of R Loops and a Regulator of RNaseH1. *Mol Cell* **65**, 832-847 e834, doi:10.1016/j.molcel.2017.01.029 (2017).
- 4 Ohle, C. *et al.* Transient RNA-DNA Hybrids Are Required for Efficient Double-Strand Break Repair. *Cell* **167**, 1001-1013 e1007, doi:10.1016/j.cell.2016.10.001 (2016).
- 5 Huang, L., Kim, Y., Turchi, J. J. & Bambara, R. A. Structure-specific cleavage of the RNA primer from Okazaki fragments by calf thymus RNase HI. *J Biol Chem* **269**, 25922-25927 (1994).
- 6 Sanz, L. A. *et al.* Prevalent, Dynamic, and Conserved R-Loop Structures Associate with Specific Epigenomic Signatures in Mammals. *Mol Cell* **63**, 167-178, doi:10.1016/j.molcel.2016.05.032 (2016).
- 7 Chen, Y. H. *et al.* Transcription shapes DNA replication initiation and termination in human cells. *Nat Struct Mol Biol* **26**, 67-77, doi:10.1038/s41594-018-0171-0 (2019).
- 8 Prendergast, L. *et al.* The CENP-T/-W complex is a binding partner of the histone chaperone FACT. *Genes Dev* **30**, 1313-1326, doi:10.1101/gad.275073.115 (2016).

REVIEWERS' COMMENTS:

Reviewer #1 (Remarks to the Author):

The authors have largely addressed my comments. I do not think that the data in Supplementary Figure 4 address my comments about DNA damage and stress in the way it is suggested in the rebuttal. These experiments were really done in response to the question by another reviewer. However, more data have been added overall, alleviating some of my concerns.

Minor comments:

The RNase H control treatments for S9.6 staining and DRIP were performed with recombinant RNase H from *E. coli*. *E. coli* only has one RNase H, so this should be denoted RNase H, not RNase H1. Please indicate in the methods section whether control samples were mock-treated with RNaseH buffer.

Reviewer #2 (Remarks to the Author):

The authors have addressed mostly all major concerns by the reviewers by performing new set of experiments, so I recommend accepting this manuscript for publication. However, this reviewer would strongly recommend mentioning the number of repeats performed for each DNA fiber experiment and using more appropriate statistical tests for all DNA fiber experiments. One-way ANOVA or Kruskal-Wallis tests with the appropriate post-test are preferred when comparing multiple non-parametric populations.

Reviewer #3 (Remarks to the Author):

This manuscript has been much improved with the addition of the new data.

There are a few minor points that need to be addressed:

1. Supplementary Fig 5b has not been labelled.
2. Supplementary Fig 7 does not illustrate what is mentioned in the relevant text (p13)- it looks like the wrong figure has been used.
3. Figure 5c the legend is in % but the y axis is expressed as a fraction. The maximum value now looks to be 0.5% rather than ~50%.

Reviewer #4 (Remarks to the Author):

The authors present good quality imaging data with extensive quantifications. Controls that were initially missing were introduced in response to reviewers' comments (for example s9.6 staining upon RNaseH1 treatment - now in Supplementary Material).

A couple of specific comments:

Colocalization studies could be improved by calculating Pearson coefficient (or similar global quantifications).

Colocalization studies using STED microscopy, given its extremely high resolution, could potentially be improved by checking (and correcting if needed) the chromatic aberration. Chromatic aberration can cause a small shift between the red and the green channels appearing as only partial overlap despite the signal originating from the same spot.

Supplementary figure 4a - some regions of interest seem to have been enhanced, giving them an artificial appearance.

References to Suppl. Figure 7 in the main text body are incorrect

I wish the authors commented a little bit more on the cytoplasmic signal in S9.6 staining. Having said that, for quantifications the authors do the right thing by masking the region of interest using DAPI channel.

Overall, however, I think that presented microscopy data are sound (especially with the controls added). Whether the interpretation of the above data is correct needs to be assessed by specialist in the field.

Response to Reviewers

REVIEWERS' COMMENTS:

Reviewer #1 (Remarks to the Author):

R1. The authors have largely addressed my comments. I do not think that the data in Supplementary Figure 4 address my comments about DNA damage and stress in the way it is suggested in the rebuttal. These experiments were really done in response to the question by another reviewer. However, more data have been added overall, alleviating some of my concerns.

Minor comments:

The RNase H control treatments for S9.6 staining and DRIP were performed with recombinant RNase H from *E. coli*. *E. coli* only has one RNase H, so this should be denoted RNase H, not RNase H1. Please indicate in the methods section whether control samples were mock-treated with RNaseH buffer.

Reply:

- The *E. coli* recombinant RNase H has been corrected in the revised version.
- The sentence "Control samples were mock-treated with RNaseH buffer" is now indicated in the Methods section.

Reviewer #2 (Remarks to the Author):

R2. The authors have addressed mostly all major concerns by the reviewers by performing new set of experiments, so I recommend accepting this manuscript for publication.

However, this reviewer would strongly recommend mentioning the number of repeats performed for each DNA fiber experiment and using more appropriate statistical tests for all DNA fiber experiments. One-way ANOVA or Kruskal-Wallis tests with the appropriate post-test are preferred when comparing multiple non-parametric populations.

Reply:

The number of repeats are now mentioned in the text to each fibre analysis figure. Additionally, in the text to figures we have included the result of Kruskal-Wallis test and in a separate table we provide the results of the Kruskal-Wallis tests for each fibre experiment, along with Dunn's multiple comparisons test.

Reviewer #3 (Remarks to the Author):

R3. This manuscript has been much improved with the addition of the new data.

There are a few minor points that need to be addressed:

1. Supplementary Fig 5b has not been labelled.

2. Supplementary Fig 7 does not illustrate what is mentioned in the relevant text (p13)- it looks like the wrong figure has been used.
3. Figure 5c the legend is in % but the y axis is expressed as a fraction. The maximum value now looks to be 0.5% rather than ~50%.

Reply:

1. We apologize for the omission. Supplementary Fig 5b has been labelled in the new version.
2. We apologize for the inadvertent mistake. The references in the text have been corrected in the revised version.
3. The y axis has now been corrected to represent % values.

Reviewer #4 (Remarks to the Author):

R4. The authors present good quality imaging data with extensive quantifications. Controls that were initially missing were introduced in response to reviewers' comments (for example s9.6 staining upon RNaseH1 treatment - now in Supplementary Material).

A couple of specific comments:

Colocalization studies could be improved by calculating Pearson coefficient (or similar global quantifications).

Reply:

The Pearson correlation coefficient for the colocalisation studies is already presented in Supplementary table 1.

R4. Colocalization studies using STED microscopy, given its extremely high resolution, could potentially be improved by checking (and correcting if needed) the chromatic aberration. Chromatic aberration can cause a small shift between the red and the green channels appearing as only partial overlap despite the signal originating from the same spot.

Reply:

The chromatic aberration was monitored and corrected during image acquisition. The partial overlap is likely resulting from a biological factor, as the overlap is not consistently observed in one direction as would be expected for mismatches caused by chromatin aberration.

R4. Supplementary figure 4a - some regions of interest seem to have been enhanced, giving them an artificial appearance.

Reply:

The enhancement was to increase the weak signal so it could be seen in the figure. The enhancements for visualisation were applied equally, across all images in the displayed

set.

R4. References to Suppl. Figure 7 in the main text body are incorrect.

Reply:

The references in the text have been corrected.

R4. I wish the authors commented a little bit more on the cytoplasmic signal in S9.6 staining. Having said that, for quantifications the authors do the right thing by masking the region of interest using DAPI channel.

Overall, however, I think that presented microscopy data are sound (especially with the controls added). Whether the interpretation of the above data is correct needs to be assessed by specialist in the field.